# On Memorization in Diffusion Models

**Xiangming Gu**[*]                                    *xiangming@u.nus.edu*
*National University of Singapore*

**Chao Du**[†]                                    *duchao@sea.com*
*Sea AI Lab*

**Tianyu Pang**[†]                                    *tianyupang@sea.com*
*Sea AI Lab*

**Chongxuan Li**                                    *chongxuanli@ruc.edu.cn*
*Gaoling School of Artificial Intelligence*
*Renmin University of China*

**Min Lin**                                    *linmin@sea.com*
*Sea AI Lab*

**Ye Wang**[†]                                    *wangye@comp.nus.edu.sg*
*National University of Singapore*

**Reviewed on OpenReview:** *https://openreview.net/forum?id=D3DBqvSDbj*

## Abstract

Due to their capacity to generate novel and high-quality samples, diffusion models have attracted significant research interest in recent years. Notably, the typical training objective of diffusion models, i.e., denoising score matching, has a closed-form optimal solution that can only generate training-data replicating samples. This indicates that a memorization behavior is theoretically expected, which contradicts the common generalization ability of state-of-the-art diffusion models, and thus calls for a deeper understanding. Looking into this, we first observe that memorization behaviors tend to occur on smaller-sized datasets, which motivates our definition of effective model memorization (EMM), a metric measuring the maximum size of training data at which a model approximates its theoretical optimum. Then, we quantify the impact of the influential factors on these memorization behaviors in terms of EMM, focusing primarily on data distribution, model configuration, and training procedure. Besides comprehensive empirical results identifying the influential factors, we surprisingly find that conditioning training data on uninformative random labels can significantly trigger the memorization in diffusion models. Our study holds practical significance for diffusion model users and offers clues to theoretical research in deep generative models. Code is available at `https://github.com/sail-sg/DiffMemorize`.

## 1 Introduction

In the last few years, diffusion models (Sohl-Dickstein et al., 2015; Song & Ermon, 2019; Ho et al., 2020; Song et al., 2021b) have achieved significant success across diverse domains of generative modeling, including image generation (Dhariwal & Nichol, 2021; Karras et al., 2022), text-to-image synthesis (Rombach et al., 2022; Ramesh et al., 2022), audio / speech synthesis (Kim et al., 2022; Huang et al., 2023), graph generation (Xu et al., 2022; Vignac et al., 2022), and 3D content generation (Poole et al., 2023; Lin et al., 2023). Substantial

---

[*]Work done during an internship at Sea AI Lab.
[†]Corresponding authors.

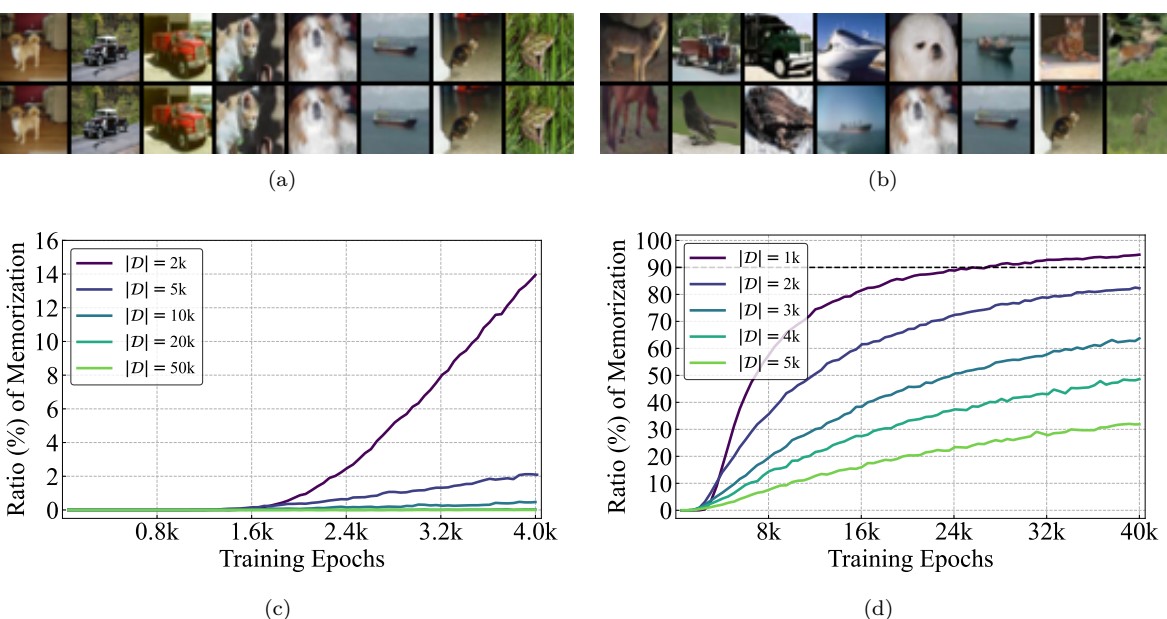

Figure 1: Overall motivation. Generated images (*top row*) and their $\ell_2$-nearest training samples in $\mathcal{D}$ (*bottom row*) by (a) the theoretical optimum defined in equation 2; (b) EDM (Karras et al., 2022). **Memorization Ratios (%)** of EDM models trained with different $|\mathcal{D}|$; (c) within 4k training epochs; (d) when extending to 40k training epochs.

empirical evidence attests to the ability of diffusion models to generate diverse and novel high-quality samples (Dhariwal & Nichol, 2021; Nichol & Dhariwal, 2021; Nichol et al., 2021), underscoring their powerful capability of abstracting and comprehending the characteristics of the training data.

Diffusion models posit a forward diffusion process $\{z_t\}_t$ ($t \in [0, T]$) that gradually introduces Gaussian noise to a data point $x$, resulting in a transition distribution $q(z_t|x) = \mathcal{N}(z_t|\alpha_t x, \sigma_t^2 \mathbf{I})$. The coefficients $\alpha_t$ and $\sigma_t$ are chosen such that the initial distribution $q_0(z_0)$ aligns with the data distribution $\mathbf{P}(x)$ while steering it towards an approximately Gaussian distribution $q_T(z_T)$. Sampling from the data distribution $\mathbf{P}$ can then be achieved by reversing this process, for which a critical unknown term is the data score $\nabla_{z_t} \log q_t(z_t)$ (Song et al., 2021b). Diffusion models approximate the data scores with a score model $\boldsymbol{s}_\theta(z_t, t)$, which is typically learned via denoising score matching (DSM) (Vincent, 2011):

$$\mathcal{J}_{\text{DSM}}(\theta) \triangleq \frac{1}{2N} \sum_{n=1}^{N} \mathbb{E}_{t,\epsilon} \left\| \boldsymbol{s}_\theta(\alpha_t x_n + \sigma_t \epsilon, t) + \frac{\epsilon}{\sigma_t} \right\|_2^2, \tag{1}$$

given $\epsilon \sim \mathcal{N}(\mathbf{0}, \mathbf{I})$ and a dataset of $N$ training samples $\mathcal{D} \triangleq \{x_n | x_n \sim \mathbf{P}(x)\}_{n=1}^{N}$. Interestingly, it is not difficult to identify the optimal solution of equation 1 (assuming sufficient capacity of $\theta$, see proof in Appendix A.1):

$$\boldsymbol{s}^*(z_t, t) = \left( \sum_{n'=1}^{N} \exp\left( -\frac{\|\alpha_t x_{n'} - z_t\|_2^2}{2\sigma_t^2} \right) \right)^{-1} \cdot \sum_{n=1}^{N} \exp\left( -\frac{\|\alpha_t x_n - z_t\|_2^2}{2\sigma_t^2} \right) \frac{\alpha_t x_n - z_t}{\sigma_t^2}, \tag{2}$$

which, however, leads the reverse process towards the empirical data distribution, defined as $\widehat{\mathbf{P}}(x) = \frac{1}{N} \sum_{n=1}^{N} \delta(x - x_n)$. Consequently, the optimal score model in equation 2 can only produce samples that replicate the training data, as shown in Figure 1(a), suggesting a *memorization* behavior (van den Burg & Williams, 2021).[1] This evidently contradicts the typical generalization capability exhibited by state-of-the-art diffusion models such as EDM (Karras et al., 2022), as illustrated in Figure 1(b).

Such intriguing gap prompts inquiries into (i) the conditions under which the learned diffusion models can faithfully approximate the optimum $\boldsymbol{s}^*$ (essentially showing memorization) and (ii) the influential

---

[1]We also provide a theoretical analysis from the lens of backward process in Appendix A.2.

factors governing memorization behaviors in diffusion models. Besides a clear issue of potential adverse generalization performance (Yoon et al., 2023), it further raises a crucial concern that diffusion models trained with equation 1 might imperceptibly memorize the training data, exposing several risks such as privacy leakage (Somepalli et al., 2023b) and copyright infringement (Somepalli et al., 2023a; Zhao et al., 2023; Wang et al., 2024). For example, Carlini et al. (2023) show that it is possible to extract a few training images from Stable Diffusion (Rombach et al., 2022), substantiating a tangible hazard.

In response to these inquiries and concerns, this paper presents a comprehensive empirical study on memorization behavior in widely adopted diffusion models, including EDM (Karras et al., 2022) and Stable Diffusion (Rombach et al., 2022). We start with an analysis of EDM, noting that memorization tends to occur when trained on smaller-sized datasets, while remaining undetectable on larger datasets. This motivates our definition of *effective model memorization* (EMM), a metric quantifying the maximum number of training data points (sampled from distribution $\mathbf{P}$) at which a diffusion model $\mathcal{M}$ demonstrates the similar memorization behavior as the theoretical optimum after the training procedure $\mathcal{T}$. We then quantify the impact of critical factors on memorization in terms of EMM on the CIFAR-10, FFHQ (Karras et al., 2019), and Imagenette (Deng et al., 2009; Somepalli et al., 2023b) datasets, considering the three facets of $\mathbf{P}$, $\mathcal{M}$, and $\mathcal{T}$. Among all illuminating results, we surprisingly observe that the memorization can be triggered by conditioning training data on completely random and uninformative labels. Specifically, using such conditioning design, we show that more than 65% of samples generated by diffusion models trained on the 50k CIFAR-10 images are replicas of training data, an obvious contrast to the original 0%. Our study holds practical significance for diffusion model users and offers clues to theoretical research in deep generative models.

## 2   Memorization in Diffusion Models

We start by examining the memorization in the widely-adopted EDM (Karras et al., 2022), which is one of the state-of-the-art diffusion models for image generation. To determine whether a generated image $x$ is a memorized replica from the training data $\mathcal{D}$, we adopt the criteria introduced in Yoon et al. (2023), which considers $x$ as memorized if its $\ell_2$ distance to the nearest neighbor is smaller than $1/3$ of that to the second nearest neighbor in the training data. Here the factor $1/3$ is an empirical threshold as it accurately aligns human perception of memorization (Yoon et al., 2023). We train an EDM model on the CIFAR-10 dataset without applying data augmentation (to avoid any ambiguity regarding memorization) and evaluate the ratio of memorization among 10k generated images. Remarkably, we observe a memorization ratio of zero throughout the entire training process, as illustrated by the bottom curve in Figure 1(c).

Intuitively, we hypothesize that the default configuration of EDM lacks the essential capacity to memorize the 50k training images in CIFAR-10, which motivates our exploration into whether expected memorization behavior will manifest when reducing the training dataset size. In particular, we generate a sequence of training datasets with different sizes of {20k, 10k, 5k, 2k}, by sampling subsets from the original set of 50k CIFAR-10 training images. We follow the default EDM training procedure (Karras et al., 2022) with consistent training epochs on these smaller datasets. As shown in Figure 1(c), when $|\mathcal{D}| = 20$k or 10k, the memorization ratio remains close to zero. However, upon further reducing the training dataset size to $|\mathcal{D}| = 5$k or 2k, the EDM models exhibit noticeable memorization. This observation indicates the substantial impact of training dataset size on memorization. Additionally, we notice that the memorization ratio increases with more training epochs. To observe this, we extend the training duration to 40k epochs, ten times longer than that in Karras et al. (2022). In Figure 1(d), when $|\mathcal{D}| = 1$k, the model achieves over 90% memorization. However, even with 40k training epochs, the diffusion model still struggles to replicate a large portion of training samples when $|\mathcal{D}| = 5$k.

Based on our findings, we seek to quantify the maximum dataset size at which diffusion models demonstrate behavior similar to the theoretical optimum, which is crucial in understanding memorization behavior. To formalize this notion, considering a data distribution $\mathbf{P}$, a diffusion model configuration $\mathcal{M}$, and a training procedure $\mathcal{T}$, we introduce the concept of *effective model memorization* with the definition as follows inspired by Yoon et al. (2023).

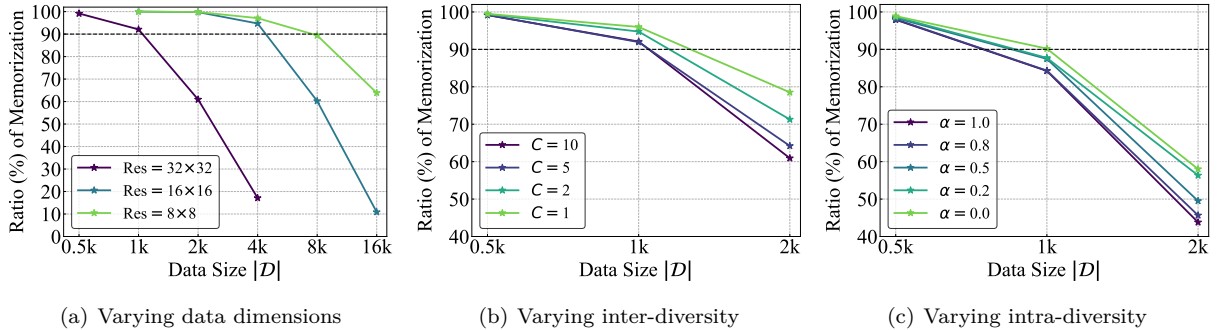

(a) Varying data dimensions      (b) Varying inter-diversity      (c) Varying intra-diversity

Figure 2: **Memorization ratios (%)** of diffusion models on CIFAR-10 under different factors of data distribution $\mathbf{P}$. The intersections of *dashed line (90%)* and different curves are the estimates of EMMs.

**Definition 2.1** (Effective model memorization). The effective model memorization (EMM) with respect to $\mathbf{P}$, $\mathcal{M}$, $\mathcal{T}$ and parameter $\zeta > 0$, is defined as

$$\mathrm{EMM}_\zeta(\mathbf{P}, \mathcal{M}, \mathcal{T}) = \max_N \left\{ \mathbb{E}[\mathcal{R}_{\mathrm{Mem}}(\mathcal{D}, \mathcal{M}, \mathcal{T})] \geq 1 - \zeta \Big| \mathcal{D} \sim \mathbf{P}, |\mathcal{D}| = N \right\}, \tag{3}$$

where $\mathcal{R}_{\mathrm{Mem}}$ refers to the ratio of memorization.

EMM indicates the condition under which the learned diffusion model approximates the theoretical optimum and reveals how $\mathbf{P}$, $\mathcal{M}$, and $\mathcal{T}$ interact and affect memorization. Our definition assumes that a higher memorization ratio tends to occur on smaller-sized training datasets, which is stated as

**Hypothesis 2.2.** Given $\mathcal{M}$, $\mathcal{T}$ and two training datasets $\mathcal{D}_1$ and $\mathcal{D}_2$, both of which are sampled from the same data distribution $\mathbf{P}$, the ratio of memorization satisfies

$$\mathcal{R}_{\mathrm{Mem}}(\mathcal{D}_1, \mathcal{M}, \mathcal{T}) \geq \mathcal{R}_{\mathrm{Mem}}(\mathcal{D}_2, \mathcal{M}, \mathcal{T}) \ \ \text{if } \mathcal{D}_1 \subset \mathcal{D}_2 \text{ and } \mathcal{D}_1, \mathcal{D}_2 \sim \mathbf{P}. \tag{4}$$

Based on Hypothesis 2.2, we provide a viable way to estimate EMM. Specifically, we sample a series of training datasets $\mathcal{D}_1, \mathcal{D}_2, ...,$ with different sizes from the data distribution $\mathbf{P}$, and then train diffusion models with configuration $\mathcal{M}$ following the training procedure $\mathcal{T}$. Afterwards, we evaluate the ratio of memorization $\mathcal{R}_{\mathrm{Mem}}(\mathcal{D}_1, \mathcal{M}, \mathcal{T}), \mathcal{R}_{\mathrm{Mem}}(\mathcal{D}_2, \mathcal{M}, \mathcal{T}), ...,$ and then determine the size of training dataset $\mathcal{D}$ which meets that $\mathcal{R}_{\mathrm{Mem}}(\mathcal{D}, \mathcal{M}, \mathcal{T}) \approx 1 - \zeta$. We note that it is computational intractable to determine the accurate value of EMM. Therefore, we interpolate the value of EMM based on two consecutive sampled datasets $\mathcal{D}_i$, $\mathcal{D}_{i+1}$ that $\mathcal{R}_{\mathrm{Mem}}(\mathcal{D}_i, \mathcal{M}, \mathcal{T}) > 1 - \zeta$ and $\mathcal{R}_{\mathrm{Mem}}(\mathcal{D}_{i+1}, \mathcal{M}, \mathcal{T}) < 1 - \zeta$. Therefore, this study is formulated as how the above three factors $\mathbf{P}$, $\mathcal{M}$, and $\mathcal{T}$ affect the measurement of EMM. There is no principled way to select the value of $\zeta$, so we set it as 0.1 based on our experiments in Figure 1(d) throughout our study. We include an analysis on the sensitivity of $\zeta$ in Appendix H.

We set CIFAR-10 as our default dataset, and the basic experimental setup is introduced in Appendix B, which is a well-adopted recipe for diffusion models. We highlight that compared to Karras et al. (2022), we run 10 times the number of training epochs. The experiments related to FFHQ (Karras et al., 2019) and Imagenette (Deng et al., 2009; Somepalli et al., 2023b) are shown in Appendix D and Section 6, respectively. For evaluation, we also consider the image quality metric, e.g., Fréchet Inception Distance (FID) (Heusel et al., 2017) in Appendix E, and alternative memorization metric in Appendix F. It is worth mentioning that when diffusion models memorize a significant proportion of training data, the metrics related to image quality (including FID) are also close to optimal. This intuition is in line with our experimental results in Appendix E. Moreover, when considering alternative memorization metrics, the relationship of various factors and memorization remains consistent.

## 3 Data Distribution P

In the preceding section, we have illustrated the substantial impact of the size of training data on the memorization in diffusion models and how we evaluate the value of effective model memorization (EMM).

We now proceed to investigate the influence of specific attributes of the data distribution $\mathbf{P}$ on the EMM, focusing primarily on the dimensions and diversity of the data. We keep both the model configuration $\mathcal{M}$ and the training procedure $\mathcal{T}$ fixed throughout this section.

## 3.1 Data Dimension

As likelihood-based generative models, diffusion models could face challenges when fitting high-resolution images, stemming from their mode-covering behavior as noted by Rombach et al. (2022). To explore the influence of data dimensionality on the memorization tendencies of diffusion models, we evaluate the EMMs on CIFAR-10 at various resolutions: 32×32, 16×16 and 8×8, where the latter two are obtained by downsampling. Note that the U-Net (Ronneberger et al., 2015) seamlessly accommodates inputs of different resolutions, requiring no modification of the model configuration $\mathcal{M}$.

For each resolution, we sample a series of training datasets $\mathcal{D}$ with varying sizes and evaluate the memorization ratios of trained diffusion models. As illustrated in Figure 2(a), we estimate the EMM for each resolution by determining the intersection between the line of 90% memorization (*dashed line*) and the memorization curve. The results reveal natural insights into the EMM with varying input dimensions. Specifically, for the 32×32 input resolution, we observe an EMM of approximately 1k. Transitioning to a 16×16 resolution, the EMM slightly surpasses 4k, while for the 8×8 resolution, it reaches approximately 8k. Furthermore, even for $|\mathcal{D}| = 16k$, the ratio of memorization still exceeds 60% when trained on 8×8 CIFAR-10 images. These results underscore the profound impact of data dimensionality on the memorization within diffusion models.

## 3.2 Data Diversity

**Number of Classes.** We consider four different data distributions by selecting $C \in \{1, 2, 5, 10\}$ classes of images from CIFAR-10 and then evaluate the EMMs. While varying the data size $|\mathcal{D}|$ during probing the EMMs, we ensure that each class contains an equal share of $|\mathcal{D}|/C$ data instances. The results of EMMs for different $C$ are shown in Figure 2(b). We find that as the number of classes increases, diffusion models tend to exhibit a lower memorization ratio and a lower EMM, which is consistent with the intuition that diverse data is harder to be memorized. Note, however, that this effect is subtle, as evidenced by the nearly identical EMMs observed for $C = 5$ and $C = 10$.

**Intra-Class Diversity.** We also explore the impact of intra-class diversity, which measures variations within individual classes. We conduct experiments with $C = 1$, where only the *Dog* class of CIFAR-10 is used. To control this diversity, we gradually blend images (scaled to a resolution of 32×32) from the *Dog* class in ImageNet (Deng et al., 2009) into the *Dog* class of CIFAR-10. We introduce an interpolation ratio, denoted as $\alpha$ ($\alpha \in [0, 1]$), representing the proportion of ImageNet data in the constructed training dataset. Notably, ImageNet's *Dog* class contains 123 sub-classes, indicating higher intra-class diversity compared to CIFAR-10. Consequently, a larger $\alpha$ corresponds to higher intra-class diversity. As shown in Figure 2(c), an increased blend of ImageNet data results in slightly lower EMM in the trained diffusion models. Similar to our experiments concerning the number of classes, these results reaffirm that diversity contributes limitedly to memorization.

## 4 Diffusion Model Configuration $\mathcal{M}$

In this section, we study the influence of different model configurations on the memorization tendencies of diffusion models. Our evaluation encompasses several aspects of model design, such as model size (width and depth), the way to incorporate time embedding, and the presence of skip connections in the U-Net. Similar to previous sections, we probe the EMMs by training multiple models on training data of different sizes from the same data distribution $\mathbf{P}$ (CIFAR-10), while keeping the training procedure $\mathcal{T}$ fixed.

## 4.1 Model Size

Diffusion models are usually constructed using the U-Net architecture (Ronneberger et al., 2015). We explore the influence of model size on memorization using two distinct approaches. First, we increase the channel multiplier, thereby augmenting the width of the model. Alternatively, we raise the number of residual blocks per resolution, as demonstrated by (Song et al., 2021b), to increase the model depth.

**Model width.** We explore different channel multipliers, specifically $\{128, 192, 256, 320\}$, while keeping the number of residual blocks per resolution fixed at 2. As illustrated in Figure 3(a), it is evident that as the

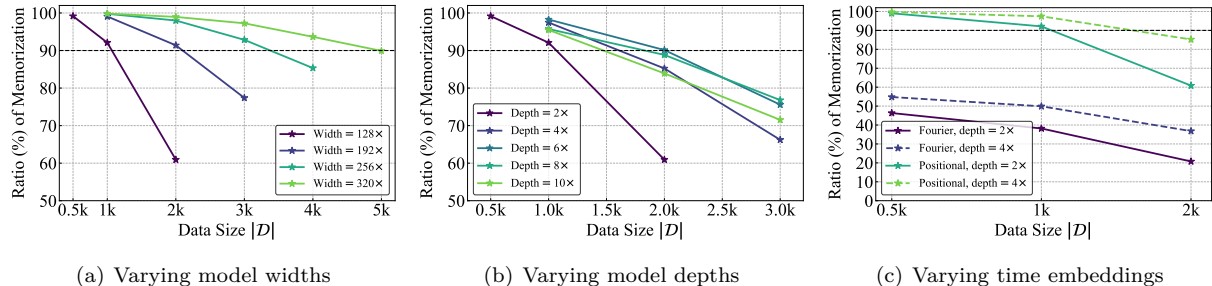

(a) Varying model widths      (b) Varying model depths      (c) Varying time embeddings

Figure 3: **Memorization ratios (%)** of diffusion models on CIFAR-10 under different factors of model configuration $\mathcal{M}$.

model width increases in diffusion models, the EMMs exhibit a monotonic rise. Notably, scaling the channel multiplier to 320 yields an EMM of approximately 5k, representing a four times increase compared to the EMM observed with a channel multiplier set at 128. These results show the direct relationship between model width and memorization in diffusion models.

**Model depth.** We vary model depth by adjusting the number of residual blocks per resolution (ranging from 2 to 12), while maintaining a constant channel multiplier of 128. In contrast to the scenario of varying model width, modifying model depth yields non-monotonic effects on memorization. As present in Figure 3(b), the EMM initially increases as we scale the number of residual blocks per resolution from 2 to 6. However, when further increasing the model depth, the EMM starts to decrease. This non-monotonic trend is further shown in Figure 7(b).

We assess the above two approaches for scaling the model size of diffusion models. Model size scales linearly when augmenting model depth but quadratically when increasing model width. With a channel multiplier set to 320, the diffusion model encompasses roughly 219M trainable parameters, yielding an EMM of approximately 5k. In contrast, with a residual block number per resolution set to 12, the model contains approximately 138M parameters, resulting in an EMM only ranging between 1k and 2k. Further increasing the model depth may encounter failures in training. Consequently, scaling model width emerges as a more viable approach for increasing the memorization ratio of diffusion models. We conduct more experiments in Appendix C.1 to further confirm our conclusions.

### 4.2 Time Embedding

In our experimental setup (see Appendix B), we employ the model architecture of DDPM++ (Song et al., 2021b; Karras et al., 2022), which incorporates positional embedding (Vaswani et al., 2017) to encode the diffusion time step. In addition to positional embedding, Song et al. (2021b) used random Fourier features (Tancik et al., 2020) in their NCSN++ models. Therefore, we conduct experiments to test both time embedding methods and assess their impact on memorization. As depicted in Figure 3(c), to further support our conclusion, we consider two diffusion models with 2 and 4 residual blocks per resolution. We observe a significant decrease in the memorization ratio (and thus EMM) when using the Fourier features in DDPM++, highlighting the noteworthy effect of time embedding on memorization. The same trend is observed on FFHQ (see Appendix D.2).

### 4.3 Skip Connections

We investigate the impact of skip connections, which are known for their significance in the success of U-Net (Ronneberger et al., 2015), on the memorization of diffusion models. Specifically, in our experimental setup (see Appendix B), the number of skip connections is $3(n+1)$, where $n$ corresponds to the number of residual blocks per resolution. For each resolution, namely, 32×32, 16×16, 8×8, we establish $n+1$ skip connections that bridge the output of the encoder to the input of the decoder. To conduct our experiments, we set the size of training dataset $|\mathcal{D}|$ to 1k, and the value of $n$ to 2, resulting in a total of 9 skip connections in our model architecture.

Initially, we explore the influence of the quantity of skip connections on memorization. Intriguingly, our observations reveal that even with a limited number of skip connections, the trained diffusion models are capable of maintaining a memorization ratio equivalent to that achieved with full skip connections, as demonstrated in

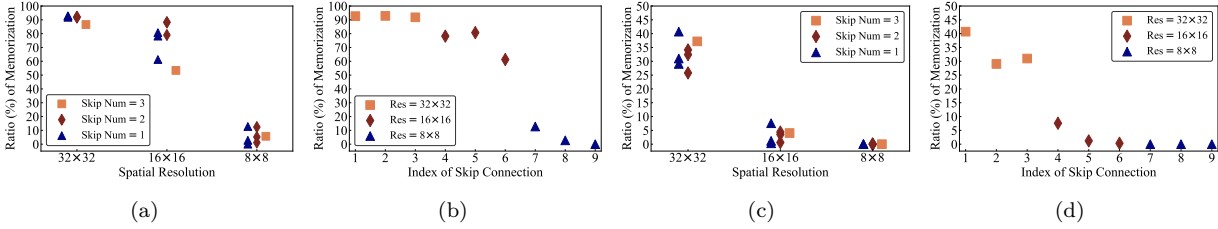

Figure 4: **Memorization ratio (%)** of diffusion models on CIFAR-10 when retaining (a) skip connections of certain spatial resolution for DDPM++; (b) single skip connection at different locations for DDPM++; (c) skip connections of certain spatial resolution for NCSN++; (d) single skip connection at different locations for NCSN++.

Appendix C.2. This underscores the role of skip connection sparsity as a significant factor influencing memorization in diffusion models, prompting our exploration of the effects of specific skip connections. In the following experiments, we consider both DDPM++ and NCSN++ architectures in Song et al. (2021b); Karras et al. (2022). Utilizing full skip connections, we observe that NCSN++ attains only approximately 45% memorization ratio, whereas DDPM++ achieves a memorization ratio exceeds 90%, all under the identical training dataset $\mathcal{D}$.

**Skip connection resolution.** We narrow our examination to the inclusion of a selected number of skip connections, specifically $m = 1, 2, 3$, all situated at a particular spatial resolution. The results, illustrated in Figure 4(a) and Figure 4(c), unveil notable trends. We note that different markers represent distinct quantities of skip connections. Our observations reveal that skip connections situated at higher resolutions contribute more significantly to memorization. Interestingly, we also find that an increase in the number of skip connections does not consistently result in higher memorization ratio. For instance, the DDPM++ model with $m = 3$ at a spatial resolution of $16 \times 16$ exhibits a lower memorization ratio compared to cases where $m = 1$ or $m = 2$.

**Skip connection location.** Additionally, we retain a single skip connection but alter its placement within the model architecture. As depicted in Figure 4(b) and Figure 4(d), with the presence of just one skip connection, DDPM++ can achieve a memorization ratio exceeding 90%, while NCSN++ attains a memorization ratio surpassing 40%, provided that this single skip connection is positioned at a resolution of $32 \times 32$. This further reinforces that skip connections at higher resolutions play a more substantial role in memorization of diffusion models.

## 5    Training Procedure $\mathcal{T}$

In Section 2, we have demonstrated that the ratio of memorization increases with the progression of training epochs, indicating the influence of training procedure $\mathcal{T}$ on memorization. Consequently, in this section, we delve into the impact of various training hyperparameters on CIFAR-10.

**Batch size.** In light of prior research highlighting the substantial role of batch size in the performance of discriminative models (Goyal et al., 2017; Keskar et al., 2016), we hypothesize that it may also influence memorization in diffusion models. Therefore, we investigate a range of batch sizes, from 128 to 896. Given that we maintain a constant number of training epochs, we apply the linear scaling rule proposed by Goyal et al. (2017) to adjust the learning rate accordingly for each batch size. Based on our basic experimental setup, we ensure a consistent ratio of learning rate to batch size $2 \times 10^{-4}/512$. As indicated in Table 2, a larger batch size correlates with a higher memorization ratio.

**Weight decay.** Weight decay is typically adopted to prevent neural networks from overfitting. Memorization can be regarded as an overfitting scenario in terms of diffusion models. Motivated by this, we explore its effect on memorization. In our basic experimental setup (see Appendix B), we follow Ho et al. (2020); Song et al. (2021b); Karras et al. (2022) to set a zero weight decay. Now we set different values of weight decay and show the memorization results in Table 2(*left*). Consequently, we find that when weight decay is ranged between $0 \sim 1 \times 10^{-3}$, it has a subtle contribution to memorization. When further increasing the value, the memorization ratio drastically decreases.

Table 1: **Memorization ratios (%)** of diffusion models on CIFAR-10 under different batch sizes.

| Batch Size | 128 | 256 | 384 | 512 | 640 | 768 | 896 |
|---|---|---|---|---|---|---|---|
| Learning rate $(10^{-4})$ | 0.5 | 1.0 | 1.5 | 2.0 | 2.5 | 3.0 | 3.5 |
| $\mathcal{R}_{\mathrm{mem}}(\%)$ $\quad |\mathcal{D}|=$1k | 89.52 | 90.87 | 91.52 | 92.09 | 91.40 | 92.05 | **92.77** |
| $\qquad\qquad\quad |\mathcal{D}|=$2k | 56.67 | 60.36 | 60.31 | 60.93 | 62.61 | **63.33** | 61.95 |

Table 2: **Memorization ratios (%)** of diffusion models on CIFAR-10 under different (*left*) weight decay values; (*right*) EMA values.

| Weight decay | $\mathcal{R}_{\mathrm{mem}}(\%)$ | |
|---|---|---|
| | $|\mathcal{D}|=$1k | $|\mathcal{D}|=$2k |
| 0 | 92.09 | 60.93 |
| $1 \times 10^{-5}$ | 91.67 | **61.63** |
| $1 \times 10^{-4}$ | **92.47** | 61.03 |
| $1 \times 10^{-3}$ | 92.11 | 58.39 |
| $1 \times 10^{-2}$ | 89.07 | 35.88 |
| $5 \times 10^{-2}$ | 13.79 | 0.03 |
| $1 \times 10^{-1}$ | 1.33 | 0.00 |

| EMA | $\mathcal{R}_{\mathrm{mem}}(\%)$ | |
|---|---|---|
| | $|\mathcal{D}|=$1k | $|\mathcal{D}|=$2k |
| 0.99929 | **92.09** | 60.93 |
| 0.999 | 91.72 | **61.38** |
| 0.99 | 91.45 | 59.27 |
| 0.9 | 90.16 | 58.31 |
| 0.5 | 90.42 | 57.80 |
| 0.1 | 90.78 | 58.00 |
| 0.0 | 90.19 | 59.20 |

**Exponential model average.** EMA was shown to effectively stabilize FIDs (Heusel et al., 2017) and remove artifacts in generated samples (Song & Ermon, 2020), as also shown in Table 6(*right*). It is widely adopted in current diffusion models (Ho et al., 2020; Dhariwal & Nichol, 2021; Karras et al., 2022). Motivated by this, we explore its effect on memorization. Previously, we fix the EMA rate as 0.99929 after the warmup following Karras et al. (2022). As present in Table 2(*right*), we investigate the memorization of diffusion models with varying EMA rates, from 0.0 to 0.99929. Although EMA values are significant for FIDs, they contribute limitedly to memorization.

## 6  Unconditional v.s. Conditional Generation

It has shown that conditional diffusion models typically yield lower FIDs than their unconditional counterparts (Karras et al., 2022). This observation motivates us to investigate whether the input condition also exerts an influence on the memorization of diffusion models. It is worth noting that to train conditional diffusion models, we slightly change $\mathcal{M}$ by incorporating a class embedding layer and adjust $\mathcal{T}$ through modifications of the training objective. We show the training objective of class-conditioned diffusion models and its optimal solution in Appendix A.3. The experiments are conducted using both EDM (Karras et al., 2022) and Stable Diffusion (Rombach et al., 2022).

**Class conditioning.** As depicted in Figure 5(a), we note that class-conditional diffusion models exhibit higher memorization ratios compared to their unconditional counterparts (thus larger EMMs). We hypothesize that this observation is attributed to the additional information introduced by these true class labels. To validate the hypothesis put forth above, we substitute the true class labels with random labels. Intriguingly, we find that the memorization of diffusion models with random labels remains at a similar level to that of models with true labels. This contradicts our previous hypothesis, as random labels are uninformative regarding training images. These results align with earlier research in the realm of discriminative models, e.g., Zhang et al. (2017), which suggests that deep neural networks can memorize training data even with randomly assigned labels. It is worth noting that when monitoring the FIDs in Figure 15, we observe that diffusion models conditioned on true labels yield lower FIDs than that on random labels. We hypothesize that with true labels as conditioning, similar training samples tend to share the same "individual model" for each class, resulting in better "individual models".

**Number of classes.** When employing random labels as conditions, the class number $C$ is not restricted to 10 in CIFAR-10. Therefore, we manipulate the choices of $C$ for random labels and then observe their effects on memorization. Initially, we compare Figure 5(a) and Figure 5(b), revealing that unconditional diffusion models and conditional models with $C=1$ maintain similar memorization ratios. However, a discernible trend emerges as we introduce more classes, with diffusion models exhibiting increased memorization (also observed on FFHQ

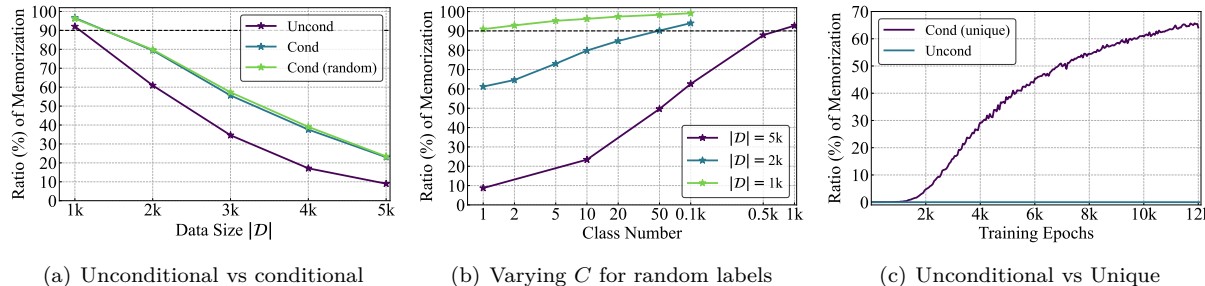

(a) Unconditional vs conditional     (b) Varying $C$ for random labels     (c) Unconditional vs Unique

Figure 5: **Memorization ratios (%)** of (a) unconditional diffusion models and conditional diffusion models on CIFAR-10 with true / random labels; (b) conditional diffusion models with $C$ random labels; (c) conditional EDM with unique labels and unconditional EDM at $|\mathcal{D}| = 50$k during the training.

in Figure 11(c)). Intriguingly, with a size of training dataset $|\mathcal{D}| = 5$k, even starting at a modest 10% memorization ratio for $C = 1$, conditional diffusion models can attain over 90% memorization ratio with $C = 1$k random labels. Based on these observations, we conclude that the number of classes significantly influences EMMs.

**Unique class condition.** We consider an extreme scenario where each training sample in $\mathcal{D}$ is assigned a unique class label, which can be regarded as the case of $C = |\mathcal{D}|$. We compare the memorization dynamics of this unique class conditional scenario with true class conditional and unconditional ones during the training, as illustrated in Figure 9. Notably, within only 8k epochs, the diffusion model with unique labels achieves a memorization ratio of more than 95%. On the FFHQ dataset, the memorization ratios of unique class conditional diffusion models are also much higher than that of unconditional ones, as present in Figure 11(d). Inspired by this, we extend this condition mechanism to encompass the entire CIFAR-10 dataset, which contains $|\mathcal{D}| = 50$k samples. It is worth noting that we train an EDM model Karras et al. (2022) to facilitate a comparison with our initial observations in Section 2. As illustrated in Figure 5(c), the unconditional EDM maintains a memorization ratio of zero throughout the training process even extending it to 12k training epochs. However, upon conditioning on the unique labels, we observe a notable shift on memorization. The trained conditional EDM achieves more than 65% memorization ratio within 12k training epochs, a significant increase compared to its previous performance. We also visualize the generated images by these two models in Appendix C.3 to further validate their memorization behaviors. These results imply that with unique labels, the training samples become strongly associated with their input conditions, rendering them more readily accessible during the generation process when the same conditions are applied.

**Extension to text condition.** Beyond unconditional diffusion models and class-conditioned diffusion models, we incorporate the experiments on state-of-the-art text-to-image diffusion models, Stable Diffusion (SD) (Rombach et al., 2022). As it is infeasibly computationally expensive to train Stable Diffusion from scratch, we opt to fine-tune the U-Net in SD on the Imagenette dataset[2], which is a realistic scenario. This dataset consists of $C = 10$ classes from the ImageNet dataset (Deng et al., 2009) and has also been adopted in Somepalli et al. (2023b). We consider two types of text prompts as conditions: (1) *plain conditioning*: using "a picture" as text prompt for all images; (2) *class conditioning*: incorporating class labels in the text prompt, i.e., "a picture of <CLASS_NAME>". For fine-tuning, we consider both full fine-tuning and LoRA fine-tuning with different ranks (Hu et al., 2021). More experimental setups are detailed in Appendix G. As shown in Table 3, we compare two distinct types of text prompts conditioning under training data with different sizes $|\mathcal{D}|$. Firstly, we observe that although SD was pretrained on billions of images, it is still prone to memorize training data when fully fine-tuning it on customized data. While LoRA fine-tuning has less tendency to memorize training data. Additionally, we find that with higher LoRA rank, SD models demonstrate higher memorization ratios. Secondly, it is noticed that with the increase of $|\mathcal{D}|$, the memorization ratio drops. Furthermore, the memorization ratios of SDs fine-tuned on class conditioning data is much higher than that on plain conditioning data. To explain this, class conditioning is similar to the case of a number of classes $C = 10$ in the class-conditioned diffusion model while plain conditioning is similar to the case $C = 1$. Our results on Stable Diffusion reaffirm the significant influence of class number on memorization within diffusion models.

---

[2]https://github.com/fastai/imagenette

Table 3: **Memorization ratio (%)** of **Stable Diffusion (SD)** on Imagenette with different fine-tuning approaches on the Imagenette dataset under two different text conditioning types.

| Fine-tuning | Condition | $|\mathcal{D}|=400$ | $|\mathcal{D}|=500$ | $|\mathcal{D}|=800$ | $|\mathcal{D}|=1000$ | $|\mathcal{D}|=2000$ |
|---|---|---|---|---|---|---|
| LoRA $r=16$ | plain | 0.00 | 0.00 | 0.00 | 0.00 | 0.00 |
|  | class | 0.00 | 0.00 | 0.00 | 0.00 | 0.00 |
| LoRA $r=256$ | plain | 0.08 | 0.01 | 0.00 | 0.00 | 0.00 |
|  | class | 5.22 | 0.25 | 0.00 | 0.00 | 0.00 |
| LoRA $r=1024$ | plain | 0.76 | 0.20 | 0.00 | 0.00 | 0.00 |
|  | class | 8.77 | 2.07 | 0.04 | 0.00 | 0.00 |
| Full fine-tuning | plain | 21.95 | 20.63 | 7.30 | 2.85 | 0.06 |
|  | class | 46.89 | 38.79 | 20.46 | 10.67 | 0.21 |

## 7 Related Work

**Memorization in discriminative models.** For discriminative models, memorization and generalization have interleaving connections. Zhang et al. (2017) first demonstrated that deep learning models can memorize the training data, even with random labeling, but generalize well. Additionally, Feldman (2020); Feldman & Zhang (2020) showed that this memorization is necessary for achieving close-to-optimal generalization under long-tailed assumption of data distribution. In the follow-up works, Baldock et al. (2021); Stephenson et al. (2021) showed that memorization predominately happens in the deep layers while Maini et al. (2023) argued that memorization is confined to a few neurons across various layers of deep neural networks. Although discriminative models can largely memorize training data, this phenomenon does not apply to diffusion models. For instance, Zhang et al. (2017) showed that the Inception v3 model (Szegedy et al., 2016) with under 25M trainable parameters can almost memorize the ImageNet dataset (Deng et al., 2009) with approximately 1.28M images. However, the EDM model (about 56M parameters) (Karras et al., 2022) can not memorize even the CIFAR-10 dataset with 50k images as present in Section 2. From another view, Bartlett et al. (2020); Nakkiran et al. (2020) showed that over-parameterized models generalize well to real data distribution and even perfectly fit to the training dataset, which is called *benign overfitting*. Nevertheless, diffusion models demonstrate adverse generalization performance (Yoon et al., 2023).

**Memorization in generative models.** Somepalli et al. (2023a); Carlini et al. (2023) represent the initial explorations on memorization in diffusion models. They investigated a range of diffusion models and demonstrated that these models memorize a few training samples and replicate them globally or in the object level during the generation. For instance, Carlini et al. (2023) identified only 50 memorized training images from 175 million generated images by Stable Diffusion (Rombach et al., 2022) and extracted 200-300 training images from $2^{20}$ generated images by DDPM and its variant (Ho et al., 2020; Nichol & Dhariwal, 2021). This highlights the memorization gap between empirical diffusion models and the theoretical optimum defined in equation 2. Somepalli et al. (2023a;b) showed that training data duplication and text conditioning play significant roles in the memorization of diffusion models from empirical studies. Another work (Wen et al., 2023) introduced a novel approach to detect memorized prompts and mitigate memorization for Stable Diffusion (Rombach et al., 2022). However, these conclusions are mostly derived from text-to-image diffusion models. The nature of memorization in diffusion models, especially for unconditional ones, remains unexplored. One recent paper Hintersdorf et al. (2024) localized the memorization to the specific neurons in text-to-image diffusion models. Another recent paper Liu et al. (2024) proposed ensembling model parameters and discarding samples with small loss values in diffusion model training, thus mitigating the memorization. Apart from diffusion models, several studies are researching towards memorization in Generative Adversarial Networks (GANs) (Webster et al., 2021; Feng et al., 2021), and language models (Carlini et al., 2021; 2022).

**Generalization in diffusion models.** Yoon et al. (2023); Kadkhodaie et al. (2023) analyzed the relationship between memorization and generalization. Yoon et al. (2023) hypothesized that generalization of diffusion models is a failure to memorize the training data when data size is large and model width is comparatively small and this memorization-generalization dichotomy may manifest at the level of classes. Our work focuses on the timing of occurrence and the influential factors regarding memorization through a much more thorough quanti-

tative analysis. Kadkhodaie et al. (2023) also showed that diffusion models memorize samples when trained on small data. However, it focused on examining the inductive bias of DNN denoisers and showed that generalization arises from the geometry-adaptive harmonic basis (GAHB) using mathematical formulations. Additionally, Zhang et al. (2023) showed that given the same noise input, different diffusion models often generate remarkably similar samples through a deterministic sampler, manifesting in memorization and generalization regimes. This provides a new perspective on understanding memorization and generalization of diffusion models. One recent paper (Kamb & Ganguli, 2024) shares motivations similar to ours. Kamb & Ganguli (2024) identified two inductive biases (locality and equivariance) in common architectures of diffusion models that prevent convergence to the ideal score function. Building on this insight, they theoretically derive equivariant local score (ELS) machines, which mirror the formulation of the ideal score function while exhibiting the generalization behavior of empirically trained models. However, these two inductive biases do not fully capture the behavior of trained models, particularly those with self-attention architectures. Analytical characterization of the exact properties of trained diffusion models remains challenging. Our work, on the empirical side, investigates how data, model designs, training processes, and conditioning prevent trained models from converging to the ideal score function.

## 8 Conclusion

In this study, we first showed that the theoretical optimum of diffusion models can only replicate training data, representing a memorization behavior. This contracts the typical generalization ability demonstrated by state-of-the-art diffusion models. To understand the memorization gap, we found that when trained on smaller-sized datasets, learned diffusion models tend to approximate the theoretical optimum. Motivated by this, we defined the notion of effective model memorization (EMM) to quantify this memorization behavior. Afterwards, we explored the impact of critical factors on memorization through the lens of EMM, from the three facets of data distribution, model configuration, and training procedure. We found that data dimension, model size, time embedding, skip connections, and class conditions play significant roles on memorization. Among all illuminating results, the memorization of diffusion models can be triggered by conditioning training data on completely random and uninformative labels. Intriguingly, when incorporating such conditioning design, more than 65% of samples generated by diffusion models trained on the entire 50k CIFAR-10 images are replicas of training data, in contrast to the original 0%. We believe that our study deepens the understanding of memorization in diffusion models and offers clues to theoretical research in the area of generative modeling.

### Acknowledgments

We would like to thank our action editor Kamalika Chaudhuri and the reviewers for their detailed and constructive comments and feedback.

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

# A  Optimal Solution of Diffusion Models

## A.1  Derivation of the Theoretical Optimum

In this section, we prove the close form of the optimal score model defined in equation 2. Firstly, the empirical denoising score matching (DSM) objective (Vincent, 2011) is:

$$\mathcal{J}_{\text{DSM}}(\theta) \triangleq \frac{1}{2N} \sum_{n=1}^{N} \mathbb{E}_{t\sim[0,T]} \mathbb{E}_{\epsilon\sim\mathcal{N}(\mathbf{0},\mathbf{I})} \left[ \lambda(t) \left\| \boldsymbol{s}_\theta(z_t, t) + \frac{\epsilon}{\sigma_t} \right\|_2^2 \right] \tag{5}$$

$$= \mathbb{E}_{t\sim[0,T]} \lambda(t) \mathbb{E}_{\epsilon\sim\mathcal{N}(\mathbf{0},\mathbf{I})} \left[ \frac{1}{2N} \sum_{n=1}^{N} \left\| \boldsymbol{s}_\theta(z_t, t) + \frac{\epsilon}{\sigma_t} \right\|_2^2 \right] \tag{6}$$

$$= \mathbb{E}_{t\sim[0,T]} \left[ \lambda(t) \int \frac{1}{2N} \sum_{n=1}^{N} \left\| \boldsymbol{s}_\theta(z_t, t) + \frac{\epsilon}{\sigma_t} \right\|_2^2 \mathcal{N}(\epsilon; \mathbf{0}, \mathbf{I}) d\epsilon \right]. \tag{7}$$

Compared to equation 1, we add a positive weighting function $\lambda(t) > 0$, which is normally used in the training of diffusion models (Song et al., 2021b). Since $z_t = \alpha_t x_n + \sigma_t \epsilon$, we have $\epsilon = -\frac{\alpha_t x_n - z_t}{\sigma_t}$. Therefore, the derivative of $\epsilon$ w.r.t. $z_t$ is $d\epsilon = \frac{dz_t}{\sigma_t}$. Then

$$\mathcal{J}_{\text{DSM}}(\theta) = \mathbb{E}_{t\sim[0,T]} \left[ \lambda(t) \int \frac{1}{2N} \sum_{n=1}^{N} \left\| \boldsymbol{s}_\theta(z_t, t) - \frac{\alpha_t x_n - z_t}{\sigma_t^2} \right\|_2^2 \mathcal{N}(z_t; \alpha_t x_n, \sigma_t^2 \mathbf{I}) \sigma_t dz_t \right] \tag{8}$$

$$= \mathbb{E}_{t\sim[0,T]} \left[ \lambda(t) \int \mathcal{J}_{\text{DSM}}(\theta, z_t, t) dz_t \right]. \tag{9}$$

To minimize the empirical DSM objective $\mathcal{J}_{\text{DSM}}(\theta)$, we can minimize $\mathcal{J}_{\text{DSM}}(\theta, z_t, t)$ given each $z_t$ since $\lambda(t) > 0$. The minimization of $\mathcal{J}_{\text{DSM}}(\theta, z_t, t)$ is a convex optimization problem w.r.t. the score $\boldsymbol{s}_\theta(z_t, t)$, which can be solved by taking the gradient w.r.t. $\boldsymbol{s}_\theta(z_t, t)$:

$$\mathbf{0} = \nabla_{\boldsymbol{s}_\theta(z_t, t)} \left[ \frac{1}{2N} \sum_{n=1}^{N} \left\| \boldsymbol{s}_\theta(z_t, t) - \frac{\alpha_t x_n - z_t}{\sigma_t^2} \right\|_2^2 \mathcal{N}(z_t; \alpha_t x_n, \sigma_t^2 \mathbf{I}) \sigma_t \right] \tag{10}$$

$$= \sum_{n=1}^{N} 2 \left[ \boldsymbol{s}(z_t, t) - \frac{\alpha_t x_n - z_t}{\sigma_t^2} \right] \mathcal{N}(z_t; \alpha_t x_n, \sigma_t^2 \mathbf{I}) \tag{11}$$

$$= \left[ \sum_{n=1}^{N} \mathcal{N}(z_t; \alpha_t x_n, \sigma_t^2 \mathbf{I}) \right] \boldsymbol{s}_\theta(z_t, t) - \sum_{n=1}^{N} \mathcal{N}(z_t; \alpha_t x_n, \sigma_t^2 \mathbf{I}) \frac{\alpha_t x_n - z_t}{\sigma_t^2}. \tag{12}$$

The optimal diffusion model can be written

$$\boldsymbol{s}^*(z_t, t) = \frac{\sum_{n=1}^{N} \mathcal{N}(z_t; \alpha_t x_n, \sigma_t^2 \mathbf{I}) \frac{\alpha_t x_n - z_t}{\sigma_t^2}}{\sum_{n'=1}^{N} \mathcal{N}(z_t; \alpha_t x_{n'}, \sigma_t^2 \mathbf{I})} \tag{13}$$

$$= \frac{\sum_{n=1}^{N} \exp\left( -\frac{\|\alpha_t x_n - z_t\|_2^2}{2\sigma_t^2} \right) \frac{\alpha_t x_n - z_t}{\sigma_t^2}}{\sum_{n'=1}^{N} \exp\left( -\frac{\|\alpha_t x_{n'} - z_t\|_2^2}{2\sigma_t^2} \right)} \tag{14}$$

$$= \sum_{n=1}^{N} \mathbb{S}\left( -\frac{\|\alpha_t x_n - z_t\|_2^2}{2\sigma_t^2} \right) \cdot \frac{\alpha_t x_n - z_t}{\sigma_t^2}, \tag{15}$$

where $\mathbb{S}$ refers to the softmax operation.

It is noted that when $\boldsymbol{s}_\theta(z_t, t)$ is parameterized using a neural network $\theta$, the objective in equation 5 is typically highly non-convex w.r.t. $\theta$. Therefore, achieving this theoretical optimum necessitates a model $\theta$

with sufficient capacity. Firstly, this theoretical optimum may lie outside the solution space of $\theta$'s model family. Even if the solution space encompasses the theoretical optimum, reaching it remains challenging due to the non-convex nature of the optimization in equation 5 w.r.t. model parameters $\theta$. The optimization may get stuck in the local optimum.

Given the existence of theoretical optimum, we find that the empirical DSM objective can be rewritten

$$\mathcal{J}_{\text{DSM}}(\theta) = \frac{1}{2N} \sum_{n=1}^{N} \mathbb{E}_{t,\epsilon} \left[ \lambda(t) \left\| \boldsymbol{s}_\theta(\alpha_t x_n + \sigma_t \epsilon, t) - \boldsymbol{s}^*(\alpha_t x_n + \sigma_t \epsilon, t) \right\|_2^2 \right] + C, \tag{16}$$

where $C = \mathbb{E}_t \left[ \lambda(t) \int \frac{1}{2N} \sum_{n=1}^{N} \left\| \boldsymbol{s}^*(z_t, t) - \frac{\alpha_t x_n - z_t}{\sigma_t^2} \right\|_2^2 \mathcal{N}(z_t; \alpha_t x_n, \sigma_t^2 \mathbf{I}) \sigma_t dz_t \right]$ is a constant value without involvement of the trained diffusion model $\theta$. This equivalence shows that diffusion models are trained to approximate the theoretical optimum. The proof is shown as below

$$\mathcal{J}_{\text{DSM}}(\theta) \tag{17}$$

$$=\mathbb{E}_t \left[ \lambda(t) \int \frac{1}{2N} \sum_{n=1}^{N} \left\| \boldsymbol{s}_\theta(z_t, t) - \frac{\alpha_t x_n - z_t}{\sigma_t^2} \right\|_2^2 \mathcal{N}(z_t; \alpha_t x_n, \sigma_t^2 \mathbf{I}) \sigma_t dz_t \right] \tag{18}$$

$$=\mathbb{E}_t \left[ \lambda(t) \int \frac{1}{2N} \sum_{n=1}^{N} \left\| \boldsymbol{s}_\theta(z_t, t) - \boldsymbol{s}^*(z_t, t) + \boldsymbol{s}^*(z_t, t) - \frac{\alpha_t x_n - z_t}{\sigma_t^2} \right\|_2^2 \mathcal{N}(z_t; \alpha_t x_n, \sigma_t^2 \mathbf{I}) \sigma_t dz_t \right] \tag{19}$$

$$=\mathbb{E}_t \left[ \lambda(t) \int \frac{1}{2N} \sum_{n=1}^{N} \left\| \boldsymbol{s}_\theta(z_t, t) - \boldsymbol{s}^*(z_t, t) \right\|_2^2 \mathcal{N}(z_t; \alpha_t x_n, \sigma_t^2 \mathbf{I}) \sigma_t dz_t \right] \tag{20}$$

$$-\mathbb{E}_t \left[ \lambda(t) \int \frac{1}{2N} \sum_{n=1}^{N} 2 \left( \boldsymbol{s}_\theta(z_t, t) - \boldsymbol{s}^*(z_t, t) \right) \left( \boldsymbol{s}^*(z_t, t) - \frac{\alpha_t x_n - z_t}{\sigma_t^2} \right) \mathcal{N}(z_t; \alpha_t x_n, \sigma_t^2 \mathbf{I}) \sigma_t dz_t \right]$$

$$+\mathbb{E}_t \left[ \lambda(t) \int \frac{1}{2N} \sum_{n=1}^{N} \left\| \boldsymbol{s}^*(z_t, t) - \frac{\alpha_t x_n - z_t}{\sigma_t^2} \right\|_2^2 \mathcal{N}(z_t; \alpha_t x_n, \sigma_t^2 \mathbf{I}) \sigma_t dz_t \right]$$

$$=\mathbb{E}_t \left[ \lambda(t) \int \frac{1}{2N} \sum_{n=1}^{N} \left\| \boldsymbol{s}_\theta(z_t, t) - \boldsymbol{s}^*(z_t, t) \right\|_2^2 \mathcal{N}(z_t; \alpha_t x_n, \sigma_t^2 \mathbf{I}) \sigma_t dz_t \right] + C \tag{21}$$

$$=\frac{1}{2N} \sum_{n=1}^{N} \mathbb{E}_{t,\epsilon} \left[ \lambda(t) \left\| \boldsymbol{s}_\theta(\alpha_t x_n + \sigma_t \epsilon, t) - \boldsymbol{s}^*(\alpha_t x_n + \sigma_t \epsilon, t) \right\|_2^2 \right] + C. \tag{22}$$

The above equation holds since

$$\int \frac{1}{2N} \sum_{n=1}^{N} 2 \left( \boldsymbol{s}_\theta(z_t, t) - \boldsymbol{s}^*(z_t, t) \right) \left( \boldsymbol{s}^*(z_t, t) - \frac{\alpha_t x_n - z_t}{\sigma_t^2} \right) \mathcal{N}(z_t; \alpha_t x_n, \sigma_t^2 \mathbf{I}) \sigma_t dz_t \tag{23}$$

$$=\frac{1}{N} \left( \boldsymbol{s}_\theta(z_t, t) - \boldsymbol{s}^*(z_t, t) \right) \int \sum_{n=1}^{N} \mathcal{N}(z_t; \alpha_t x_n, \sigma_t^2 \mathbf{I}) \sigma_t \left( \boldsymbol{s}^*(z_t, t) - \frac{\alpha_t x_n - z_t}{\sigma_t^2} \right) dz_t \tag{24}$$

$$=0. \tag{25}$$

In Karras et al. (2022), the authors have derived the optimal denoised function, while in Yi et al. (2023), the authors provided the closed form of the optimal DDPM (Ho et al., 2020). Therefore, we show the equivalence of our equation 2 with the above two forms.

For DDPM, which is trained under noise prediction objective, Kingma & Gao (2023) gave the transformation between the two parameterizations:

$$\boldsymbol{s}_\theta(z_t, t) = -\frac{\boldsymbol{\epsilon}_\theta(z_t, t)}{\sigma_t}. \tag{26}$$

Therefore, the optimal DDPM can be represented as

$$\boldsymbol{\epsilon}^*(z_t, t) = -\sigma_t \boldsymbol{s}^*(z_t, t) = \frac{\sum_{n=1}^{N} \mathcal{N}(z_t; \alpha_t x_n, \sigma_t^2 \mathbf{I}) \frac{z_t - \alpha_t x_n}{\sigma_t}}{\sum_{n'=1}^{N} \mathcal{N}(z_t; \alpha_t x_{n'}, \sigma_t^2 \mathbf{I})} \tag{27}$$

$$= \frac{z_t}{\sigma_t} - \frac{\alpha_t}{\sigma_t} \sum_{n=1}^{N} \frac{\exp\left(-\frac{\|\alpha_t x_n - z_t\|_2^2}{2\sigma_t^2}\right) x_n}{\sum_{n'=1}^{N} \exp\left(-\frac{\|\alpha_t x_{n'} - z_t\|_2^2}{2\sigma_t^2}\right)}. \tag{28}$$

In DDPM (Ho et al., 2020), the forward process is defined as $z_t = \sqrt{\bar{\alpha}_t} x + \sqrt{1 - \bar{\alpha}_t}\epsilon$, so the closed form for optimal DDPM is

$$\boldsymbol{\epsilon}^*(z_t, t) = \frac{z_t}{\sqrt{1 - \bar{\alpha}_t}} - \frac{\bar{\alpha}_t}{\sqrt{1 - \bar{\alpha}_t}} \sum_{n=1}^{N} \frac{\exp\left(-\frac{\|z_t - \bar{\alpha}_t x_n\|_2^2}{2(1 - \bar{\alpha}_t)}\right) x_n}{\sum_{n'=1}^{N} \exp\left(-\frac{\|z_t - \bar{\alpha}_t x_{n'}\|_2^2}{2(1 - \bar{\alpha}_t)}\right)}, \tag{29}$$

which is the same as Theorem 2 in Yi et al. (2023).

For denoised function, which is trained under data prediction objective, Kingma & Gao (2023) gave:

$$\boldsymbol{s}_\theta(z_t, t) = -\frac{z_t - \alpha_t D_\theta(z_t, t)}{\sigma_t^2}. \tag{30}$$

Therefore, the optimal denoised function can be represented as

$$D^*(z_t, t) = \frac{\sigma_t^2 \boldsymbol{s}^*(z_t, t) + z_t}{\alpha_t} = \frac{\sum_{n=1}^{N} \mathcal{N}(z_t; \alpha_t x_n, \sigma_t^2 \mathbf{I}) x_n}{\sum_{n'=1}^{N} \mathcal{N}(z_t; \alpha_t x_{n'}, \sigma_t^2 \mathbf{I})}. \tag{31}$$

In EDM (Karras et al., 2022), the forward process is defined as $z_t = x + \sigma_t \epsilon$, so the closed form for optimal denoised function is:

$$D^*(z_t, t) = \frac{\sum_{n=1}^{N} \mathcal{N}(z_t; x_n, \sigma_t^2 \mathbf{I}) x_n}{\sum_{n'=1}^{N} \mathcal{N}(z_t; x_{n'}, \sigma_t^2 \mathbf{I})}. \tag{32}$$

which is the same as Eq. (57) in Karras et al. (2022).

## A.2 Backward Process of the Optimal Diffusion Model

We analyze the memorization behavior of the optimal diffusion model $\boldsymbol{s}^*(z_t, t)$ defined in Eq. equation 2 through the lens of backward process. As shown in Kingma et al. (2021), the backward process of our defined diffusion models is governed by the following stochastic differential equation (SDE):

$$dz_t = \left[f(t)z_t - g^2(t)\boldsymbol{s}_\theta(z_t, t)\right] dt + g(t)dW_t, \tag{33}$$

where $W_t$ is a standard Brownian motion, and $f(t)$ and $g(t)$ follow

$$f(t) = \frac{d\log\alpha_t}{dt}, \quad g^2(t) = \frac{d\sigma_t^2}{dt} - 2\frac{d\log\alpha_t}{dt}\sigma_t^2. \tag{34}$$

Besides the SDE, Song et al. (2021b) showed that there exists an ordinary differential equation (ODE) for deterministic backward process

$$dz_t = \left[f(t)z_t - \frac{1}{2}g^2(t)\boldsymbol{s}_\theta(z_t, t)\right] dt. \tag{35}$$

We first show how to adopt the above ODE to generate samples using the optimal score model defined in equation 2. Specifically, we sample multiple time steps $0 = t_0 < \xi = t_1 < ... < t_n = T$, where $\xi$ refers to a

small value close to 0 and $T > 0$ represents the maximum time step. For simplicity, we consider Euler solver, and then we have the following update rule

$$z_{t_n} = z_{t_{n+1}} + \left( \frac{d \log \alpha_t}{dt} z_t - \frac{1}{2} \left( \frac{d\sigma_t^2}{dt} - 2 \frac{d \log \alpha_t}{dt} \sigma_t^2 \right) \boldsymbol{s}_\theta(z_t, t) \right) \Bigg|_{t=t_{n+1}} (t_n - t_{n+1}). \tag{36}$$

We also use Euler method to approximate $\frac{d \log \alpha_t}{dt}$ and $\frac{d\sigma_t^2}{dt}$ considering

$$\lim_{t_n - t_{n+1} \to 0^-} \frac{d \log \alpha_t}{dt} \Bigg|_{t=t_{n+1}} = \lim_{t_n - t_{n+1} \to 0^-} \frac{1}{\alpha_t} \frac{d\alpha_t}{dt} \Bigg|_{t=t_{n+1}} = \lim_{t_n - t_{n+1} \to 0^-} \frac{1}{\alpha_{t_{n+1}}} \frac{\alpha_{t_n} - \alpha_{t_{n+1}}}{t_n - t_{n+1}}, \tag{37}$$

$$\lim_{t_n - t_{n+1} \to 0^-} \frac{d\sigma_t^2}{dt} \Bigg|_{t=t_{n+1}} = \lim_{t_n - t_{n+1} \to 0^-} 2\sigma_t \frac{d\sigma_t}{dt} \Bigg|_{t=t_{n+1}} = \lim_{t_n - t_{n+1} \to 0^-} 2\sigma_{t_{n+1}} \frac{\sigma_{t_n} - \sigma_{t_{n+1}}}{t_n - t_{n+1}}. \tag{38}$$

Then we have

$$z_{t_n} = \frac{\alpha_{t_n}}{\alpha_{t_{n+1}}} z_{t_{n+1}} - \left( \sigma_{t_{n+1}}(\sigma_{t_n} - \sigma_{t_{n+1}}) - \frac{(\alpha_{t_n} - \alpha_{t_{n+1}})\sigma_{t_{n+1}}^2}{\alpha_{t_{n+1}}} \right) \boldsymbol{s}_\theta(z_{t_{n+1}}, t_{n+1}) \tag{39}$$

$$= \frac{\alpha_{t_n}}{\alpha_{t_{n+1}}} z_{t_{n+1}} - \left( \sigma_{t_{n+1}}\sigma_{t_n} - \frac{\alpha_{t_n}\sigma_{t_{n+1}}^2}{\alpha_{t_{n+1}}} \right) \boldsymbol{s}_\theta(z_{t_{n+1}}, t_{n+1}). \tag{40}$$

For $t_0 = 0$, we know that $\alpha_0 = 1$ and $\sigma_0 = 0$. $z_0$ refers to the generated samples, and we have

$$z_0 = \frac{z_\xi}{\alpha_\xi} + \frac{\sigma_\xi^2}{\alpha_\xi} \boldsymbol{s}_\theta(z_\xi, \xi) \tag{41}$$

$$= \frac{z_\xi}{\alpha_\xi} + \frac{\sigma_\xi^2}{\alpha_\xi} \sum_{n=1}^{N} \frac{\exp\left(-\frac{\|\alpha_\xi x_n - z_\xi\|_2^2}{2\sigma_\xi^2}\right)}{\sum_{n'=1}^{N} \exp\left(-\frac{\|\alpha_\xi x_{n'} - z_\xi\|_2^2}{2\sigma_\xi^2}\right)} \cdot \frac{\alpha_\xi x_n - z_\xi}{\sigma_\xi^2} \tag{42}$$

$$= \sum_{n=1}^{N} \frac{\exp\left(-\frac{\|\alpha_\xi x_n - z_\xi\|_2^2}{2\sigma_\xi^2}\right)}{\sum_{n'=1}^{N} \exp\left(-\frac{\|\alpha_\xi x_{n'} - z_\xi\|_2^2}{2\sigma_\xi^2}\right)} \cdot x_n. \tag{43}$$

From the above equation, we conclude that the generated samples by the optimal diffusion model are the linear combinations of training samples in $\mathcal{D}$.

Next, we consider a discrete distribution, and suppose $z = \frac{z_\xi}{\alpha_\xi}$, then

$$p(z = x_n, \xi) = \frac{\exp\left(-\frac{\|z - x_n\|_2^2}{2\alpha_\xi^2 \sigma_\xi^2}\right)}{\sum_{n'=1}^{N} \exp\left(-\frac{\|z - x_{n'}\|_2^2}{2\alpha_\xi^2 \sigma_\xi^2}\right)}, \quad n = 1, 2, ..., N. \tag{44}$$

Suppose $x_m = \text{NN}_1(z, \mathcal{D}) = \arg\min_{x \in \mathcal{D}} \|z - x\|_2^2$, then we have

$$\|z - x_m\|_2^2 - \|z - x_{n \neq m}\|_2^2 < 0. \tag{45}$$

When $\xi \to 0^+$, $\eta = \alpha_\xi \sigma_\xi \to 0^+$, then we have

$$\lim_{\xi \to 0^+} p(z = x_m, \xi) = \lim_{\eta \to 0^+} \frac{\exp\left(-\frac{\|z - x_m\|_2^2}{2\eta^2}\right)}{\sum_{n'=1}^N \exp\left(-\frac{\|z - x_{n'}\|_2^2}{2\eta^2}\right)} \tag{46}$$

$$= \lim_{\eta \to 0^+} \frac{1}{\sum_{n' \neq m}^N \exp\left(-\frac{\|z - x_{n'}\|_2^2 - \|z - x_m\|_2^2}{2\eta^2}\right) + 1} \tag{47}$$

$$= \frac{1}{\sum_{n' \neq m}^N \lim_{\eta \to 0^+} \exp\left(-\frac{\|z - x_{n'}\|_2^2 - \|z - x_m\|_2^2}{2\eta^2}\right) + 1} \tag{48}$$

$$= \frac{1}{\sum_{n' \neq m}^N \lim_{\eta \to +\infty} \exp\left(\frac{\eta^2}{2}\left(\|z - x_m\|_2^2 - \|z - x_{n'}\|_2^2\right)\right) + 1} \tag{49}$$

$$= 1. \tag{50}$$

Then

$$\lim_{\xi \to 0^+} \left( p(z = x_m, \xi) + \sum_{n' \neq m}^N p(z = x_{n'}, \xi) \right) = 1, \tag{51}$$

$$\sum_{n' \neq m}^N \lim_{\xi \to 0^+} p(z = x_{n'}, \xi) = 0. \tag{52}$$

Given $p(z = x_i, \xi) \geq 0$,

$$\lim_{\xi \to 0^+} p(z = x_{n \neq m}, \xi) = 0. \tag{53}$$

$$\lim_{\xi \to 0^+} z_0 = \lim_{\xi \to 0^+} \mathbb{E}_{p(z = x_n, \xi)}[z] = \mathrm{NN}_1(z, \mathcal{D}). \tag{54}$$

From the above analysis, we conclude that when $t_1 = \xi$ is closed to 0, the probabilistic ODE solver returns a training sample in $\mathcal{D}$.

Next, we consider an SDE solver, then the update rule is

$$z_{t_n} = z_{t_{n+1}} + \left(\frac{d\log\alpha_t}{dt} z_t - \left(\frac{d\sigma_t^2}{dt} - 2\frac{d\log\alpha_t}{dt}\sigma_t^2\right) s_\theta(z_t, t)\right)\Bigg|_{t=t_{n+1}} (t_n - t_{n+1}) \tag{55}$$

$$+ \sqrt{\frac{d\sigma_t^2}{dt} - 2\frac{d\log\alpha_t}{dt}\sigma_t^2}\Bigg|_{t=t_{n+1}} (t_n - t_{n+1})\epsilon$$

$$= \frac{\alpha_{t_n}}{\alpha_{t_{n+1}}} z_{t_{n+1}} - 2\left(\sigma_{t_{n+1}}\sigma_{t_n} - \frac{\alpha_{t_n}\sigma_{t_{n+1}}^2}{\alpha_{t_{n+1}}}\right) s_\theta(z_{t_{n+1}}, t_{n+1}) \tag{56}$$

$$+ \sqrt{2\left(\sigma_{t_{n+1}}\sigma_{t_n} - \frac{\alpha_{t_n}\sigma_{t_{n+1}}^2}{\alpha_{t_{n+1}}}\right)(t_n - t_{n+1})}\epsilon,$$

where $\epsilon \sim \mathcal{N}(\mathbf{0}, \mathbf{I})$ is a Gaussian noise. Similarly, we consider the update step at $t_0 = 0$

$$z_0 = \frac{z_\xi}{\alpha_\xi} + 2\frac{\sigma_\xi^2}{\alpha_\xi} s_\theta(z_\xi, \xi) + \sqrt{2\frac{\sigma_\xi^2 \xi}{\alpha_\xi}}\epsilon = 2\sum_{n=1}^N \frac{\exp\left(-\frac{\|\alpha_\xi x_n - z_\xi\|_2^2}{2\sigma_\xi^2}\right)}{\sum_{n'=1}^N \exp\left(-\frac{\|\alpha_\xi x_{n'} - z_\xi\|_2^2}{2\sigma_\xi^2}\right)} \cdot x_n - \frac{z_\xi}{\alpha_\xi} + \sqrt{2\frac{\sigma_\xi^2 \xi}{\alpha_\xi}}\epsilon. \tag{57}$$

When $\xi \to 0^+$

$$\lim_{\xi \to 0^+} z_0 = 2\mathrm{NN}_1\left(\frac{z_\xi}{\alpha_\xi}, \mathcal{D}\right) - \lim_{\xi \to 0^+} \frac{z_\xi}{\alpha_\xi}. \tag{58}$$

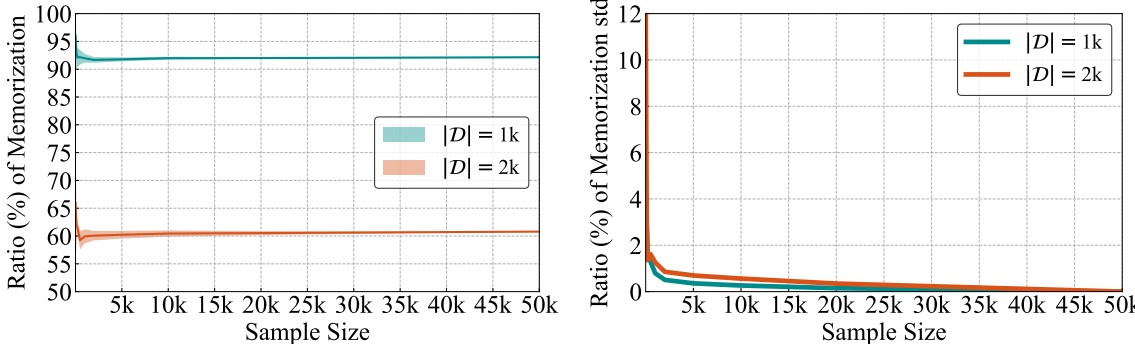

Figure 6: (*Left*) Memorization ratios (%) (*Right*) Standard deviations of memorization ratios (%) of diffusion models with different sample sizes.

Consider $\lim_{\xi \to 0^+} \frac{z_\xi}{\alpha_\xi} = \lim_{\xi \to 0^+} z_0$

$$\lim_{\xi \to 0^+} z_0 = \text{NN}_1 \left( \frac{z_\xi}{\alpha_\xi}, \mathcal{D} \right). \tag{59}$$

To summarize, through our analysis, we find that the optimal diffusion model always replicates training data through the backward process.

### A.3   Optimal Class-conditioned Diffusion Model

In the above, we provide the derivation of the optimal diffusion model for unconditional generation under the assumption of empirical data distribution $x \sim \frac{1}{N} \sum_{n=1}^{N} \delta(x - x_n)$. Next, we consider the scenario of class-conditional generation. The dataset can be represented as $\mathcal{D} \triangleq \{x_n, y_n\}_{n=1}^{N}, y_n \in \{1, 2, ..., C\}$, where $C$ is the number of classes. Then the empirical joint distribution of data and label is $x, y \sim \frac{1}{N} \sum_{n=1}^{N} \delta(x - x_n)\delta(y - y_n)$.

For class-conditional generation, the empirical DSM objective is written

$$\mathcal{J}_{\text{DSM}}(\theta) = \frac{1}{2N} \sum_{n=1}^{N} \mathbb{E}_{t,\epsilon} \left[ \lambda(t) \left\| s_\theta(\alpha_t x_n + \sigma_t \epsilon, y_n, t) + \frac{\epsilon}{\sigma_t} \right\|_2^2 \right] \tag{60}$$

$$= \frac{1}{2N} \sum_{c=1}^{C} \sum_{n=1}^{N_c} \mathbb{E}_{t,\epsilon} \left[ \lambda(t) \left\| s_\theta(\alpha_t x_n^c + \sigma_t \epsilon, c, t) + \frac{\epsilon}{\sigma_t} \right\|_2^2 \right],$$

where $x_n^c$ refers to the $n$-th sample with class label $c$, and $N_c$ represents the number of class label $c$.

Similarly, by taking the gradient w.r.t. $s_\theta(z_t, c, t)$, we derive the optimal class-conditioned diffusion model for each class condition $c$

$$s_\theta^*(z_t, c, t) = \frac{\sum_{n=1}^{N_c} \exp\left(-\frac{\|\alpha_t x_n^c - z_t\|_2^2}{2\sigma_t^2}\right) \frac{\alpha_t x_n^c - z_t}{\sigma_t^2}}{\sum_{n'=1}^{N_c} \exp\left(-\frac{\|\alpha_t x_{n'}^c - z_t\|_2^2}{2\sigma_t^2}\right)} = \sum_{n=1}^{N_c} \mathbb{S}\left(-\frac{\|\alpha_t x_n^c - z_t\|_2^2}{2\sigma_t^2}\right) \cdot \frac{\alpha_t x_n^c - z_t}{\sigma_t^2}. \tag{61}$$

## B   Implementation Details of Our Basic Experimental setup

**Data distribution.** Most of our experiments are conducted on the CIFAR-10 dataset, which consists of 50k RGB images with a spatial resolution of 32×32. CIFAR-10 has 10 classes, each of which has 5k images. We also incorporate the experiments on the FFHQ dataset (Karras et al., 2019) to emphasize our findings. The FFHQ dataset comprises 70k RGB images of human faces with an original resolution of

1024×1024. We down-sample these images to a resolution of 64×64 following Karras et al. (2022) to conduct our experiments. When modifying the intra-diversity of data distribution, we blend several images from the ImageNet dataset (Deng et al., 2009) to construct training datasets. In our study, we disable the data augmentation to prevent any ambiguity regarding memorization.

**Model configuration.** We consider the baseline VP configuration in Karras et al. (2022).[3] The model architecture is DDPM++, which is based on U-Net (Ronneberger et al., 2015). As our basic model, we select the number of residual blocks per resolution in the U-Net as 2 instead of 4 in original implementations. For diffusion models trained on CIFAR-10, the channel multiplier is 128, resulting in 256 channels at all resolutions of 32×32, 16×16, and 8×8. While for diffusion models trained on FFHQ, the employed U-Net has 128 channels at the resolution of 64×64, and 256 channels at the resolutions of 32×32, 16×16, and 8×8. The time embedding is positional encoding (Vaswani et al., 2017).

**Training procedure.** Our diffusion models are trained using Adam optimizer (Kingma & Ba, 2014) with a learning rate of $2 \times 10^{-4}$ and a batch size of 512 for CIFAR-10 while 256 for FFHQ. The training duration is 40k epochs, while it is 4k epochs in Karras et al. (2022). It is worth mentioning that for different training sizes, the number of training epochs is the same. Therefore, for smaller training datasets, the number of total training steps will be smaller. This setup ensures that the frequency of each image being drawn during the training procedure is the same. We schedule the learning rate and EMA rate similar to Karras et al. (2022) but in an epoch-wise manner. In the first 200 epochs, we warm-up the learning rate and the EMA rate with the increase of training iterations. Afterwards, the learning rate is fixed to $2 \times 10^{-4}$ and the EMA rate is fixed to 0.99929.

**Generation and evaluation.** We follow the backward process in Karras et al. (2022) to generate images to compute the memorization metric. To decide appropriate sample size, we first train two diffusion models on CIFAR-10 according to our basic experimental setup at $|\mathcal{D}| = 1$k and $|\mathcal{D}| = 2$k. Afterwards, we generate 50k images by each model and then bootstrap different number of samples to compute memorization ratios. As present in Figure 6, we find that the ratio of memorization has a negligible variance when sample size more than 10k images. Therefore, we generate 10k images throughout this study. We report the highest memorization ratio during the training process. To ensure the consistency in our analysis, we maintain the same set of 10k samples for computing FIDs and alternative memorization metrics. Specifically, we compute FIDs using 10k generated samples and the training dataset $\mathcal{D}$ (with varying sizes $|\mathcal{D}|$).

**Experimental compute resource.** All experiments were run on 8 NVIDIA A100 GPUs, each with 80GB of memory. The running time of each experiment highly depends on the model configuration and data size. For example, it takes about 120 hours to train an unconditional model or conditional model with unique labels in Figure 5(c).

**Experimental statistical significance.** We also evaluate the variance of memorization ratios across multiple trials. Specifically, on CIFAR-10, we run a pair of experiments focused on data distribution, primarily by altering the number of classes. We set the training data size as $|\mathcal{D}| = 1$k and the number of classes as $C = 2$, 5, and then execute each experiment over three distinct random seeds. Consequently, the memorization ratio for $C = 2$ stands at 94.59±0.19%, whereas for $C = 5$, it is recorded at 92.32±0.14%. It is noticed that the variance of memorization ratios is less than 0.2%, which is insignificant. Considering this minimal fluctuation and difficulty for repeating all experiments, we run each experiment once in the main paper.

## C   More Empirical Results on CIFAR-10

### C.1   Model Size

**Model width.** We investigate the influence of different channel multipliers, specifically exploring the values from {64, 96, 128, 160, 192, 224, 256}. We also consider two scenarios for the number of residual blocks per resolution: 2 and 4. As illustrated in Figure 7(a), when $|\mathcal{D}| = 2$k, we observe a consistent and monotonic increase in the memorization ratio as the model width grows. This observation aligns with the conclusions

---

[3]We run the training configuration C in Karras et al. (2022) after adjusting hyper-parameters and redistributing capacity compared to Song et al. (2021b).

drawn in Sec. 4.1. Furthermore, we provide a dynamic view of the memorization ratios during the training process in Figure 7(c) and Figure 7(e). It is worth noting that wider diffusion models consistently exhibit higher levels of memorization throughout the entire training process.

**Model depth.** We delve into the impact of varying numbers of residual blocks per resolution, considering values in the range of {2, 4, 6, 8, 10}. We maintain two different channel multiplier values, specifically 128 and 256, and set the training data size $|\mathcal{D}| = 2$k. The results, as presented in Figure 7(b), confirm the non-monotonic effect of model depth on memorization. . When the channel multiplier is set to 256, the curve depicting the relationship between memorization ratio and the number of residual blocks per resolution exhibits multiple peaks. To gain a better understanding of this non-monotonic effect, we visualize the training process in Figure 7(d) and Figure 7(f). Occasionally, deeper diffusion models yield lower memorization ratios than shallower ones throughout the whole training process. It is noteworthy that when both model width and depth are set large values (e.g., the number of residual blocks per resolution is 8 and the channel multiplier is 256), the memorization of the diffusion model also demonstrates non-monotonic characteristics.

## C.2 Skip Connections

We employ DDPM++ (Song et al., 2021b; Karras et al., 2022) to explore the influence of skip connection quantity on memorization. Specifically, we consider retaining $m = 1, 3, 5, 7, 8, 9$ skip connections. To keep the costs tractable, we randomly select five distinct combinations of skip connections for each specific $m$. Our findings, depicted in Figure 8(a), illustrate a consistent memorization ratio for larger values of $m$, whereas a considerable degree of variance is observed for smaller values of $m$. Notably, when $m = 1$, the memorization ratio exhibits substantial variability, spanning a range from approximately 0% to over 90%.

In addition to remaining specific skip connections at different resolutions in the main paper, we also investigate the effect of selectively deleting certain skip connections at varying resolutions. As present in Figure 8(b), when deleting one or two skip connections at any spatial resolution, the memorization ratios consistently remain above 90%. It is noteworthy that the memorization ratio only falls below 90% when three skip connections are deleted at the resolution of 32×32. These findings further confirm the pivotal role played by skip connections at higher resolutions on memorization of diffusion models.

## C.3 Unconditional v.s. Conditional Generation

We compare the memorization ratios of unconditional diffusion models, conditional diffusion models with true labels, and conditional diffusion models with unique labels. As summarized in Table 4, it is noticed that with unique labels as class conditions, the trained diffusion models can only replicate training data as the memorization ratios achieve close to 100% when $|\mathcal{D}| \leq 5$k. Additionally, we visualize the memorization ratios of diffusion models during the training in Figure 9, which reaffirms that the diffusion models with unique labels as input conditions memorize training data quickly. Typically, they can achieve more than 90% memorization ratios within 8k training epochs.

Finally, when trained on the entire CIFAR-10 dataset, i.e., $|\mathcal{D}| = 50$k, the memorization ratio of conditional EDM with unique labels achieves more than 65% within 12k training epochs, as present in Figure 5(c). However, its unconditional counterpart still maintains a zero value of memorization ratio. To further demonstrate this memorization gap, we visualize the generated images and their $\ell_2$-nearest training samples in $\mathcal{D}$ by the above two models in Figure 10. It is noticed that the conditional EDM with unique labels replicate a large proportion of training data while the unconditional EDM generates novel samples.

# D More Empirical Results on FFHQ

## D.1 Data Dimension

We investigate the influence of data dimension on the memorization of diffusion models, particularly those trained using the FFHQ dataset. Specifically, we evaluate various resolutions: 64×64, 32×32, 16×16, with the latter two resolutions achieved by downsampling. We keep the model configurations and training procedure the same. As shown in Figure 11(a), for the 64×64 input resolution, we observe an EMM between 500 and

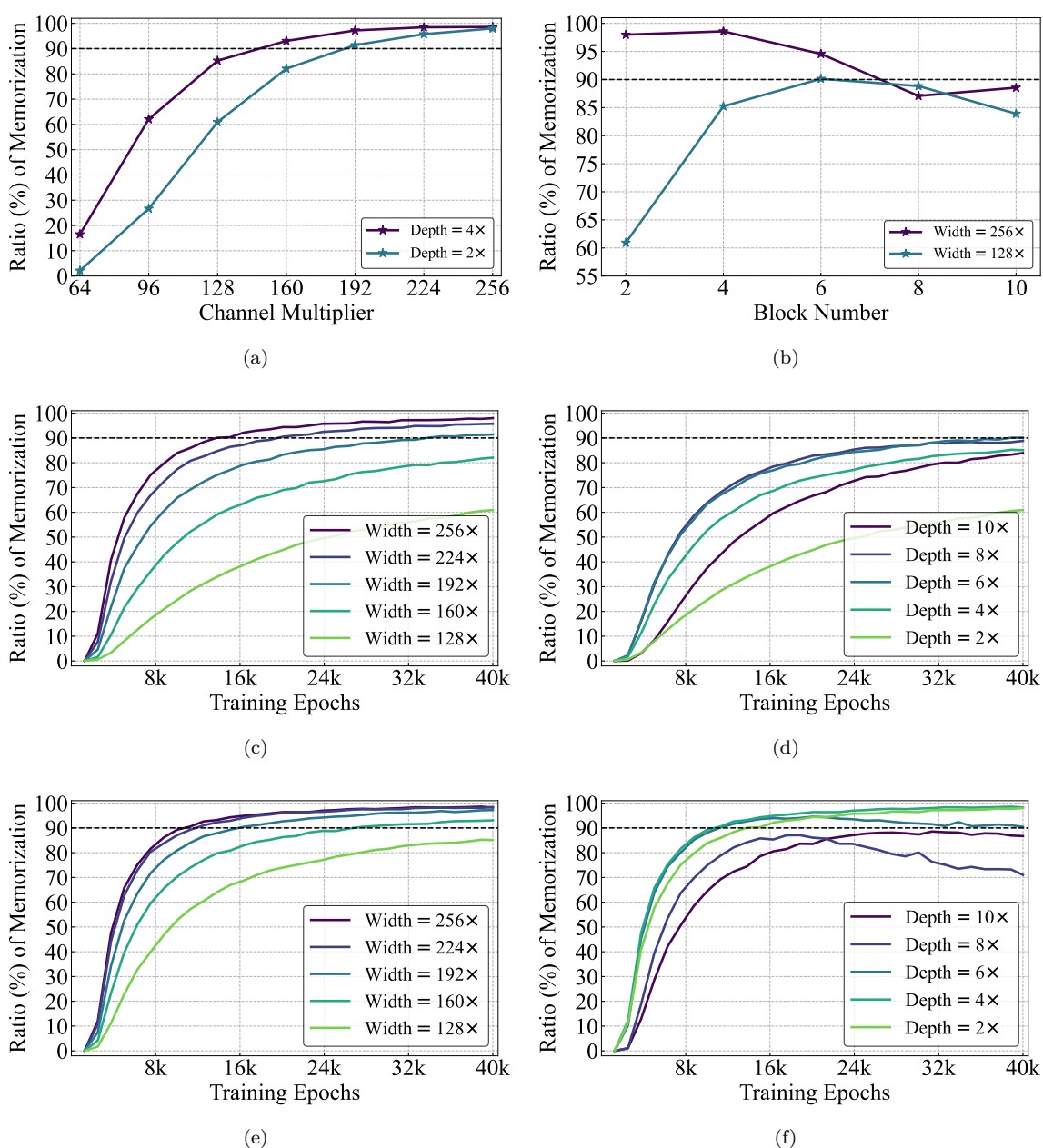

Figure 7: At the size of training data $|\mathcal{D}| = 2k$, **memorization ratio (%)** of diffusion models on CIFAR-10 with (a) varying model widths under 2 or 4 residual blocks per resolution; (b) varying model depths under the channel multiplier as 128 or 256; (c) varying model widths under 2 residual blocks per resolution during the training; (d) varying model depths under the channel multiplier as 128 during the training; (e) varying model widths under 4 residual blocks per resolution during the training; (f) varying model depths under the channel multiplier as 256 during the training.

1k. While for the 32×32 input resolution, the EMM is close to 4k. Furthermore, when input resolution is 16×16, the memorization ratio is still above 90% even for $|\mathcal{D}| = 8k$. The EMM is estimated to be between 8k and 10k. These results further indicate the significance of data dimensionality on the memorization within diffusion models.

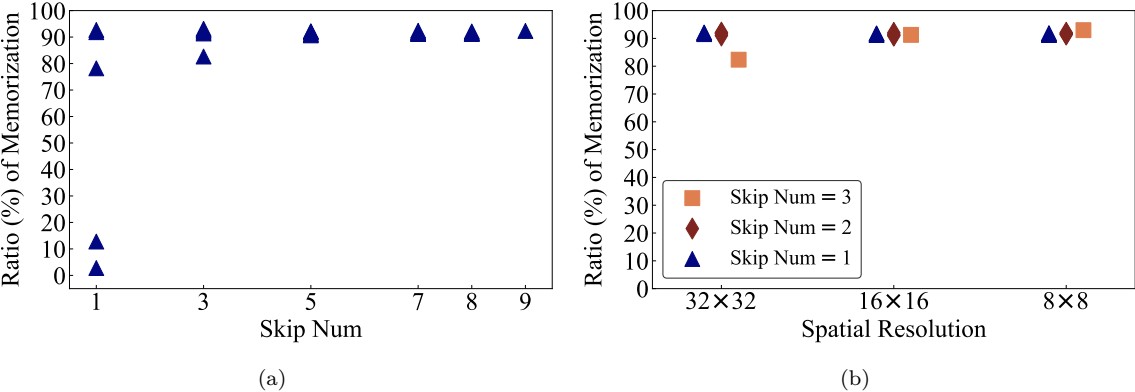

Figure 8: **Memorization ratio (%)** of diffusion models on CIFAR-10 when (a) retaining different numbers of skip connections; (b) deleting skip connections of certain spatial resolution.

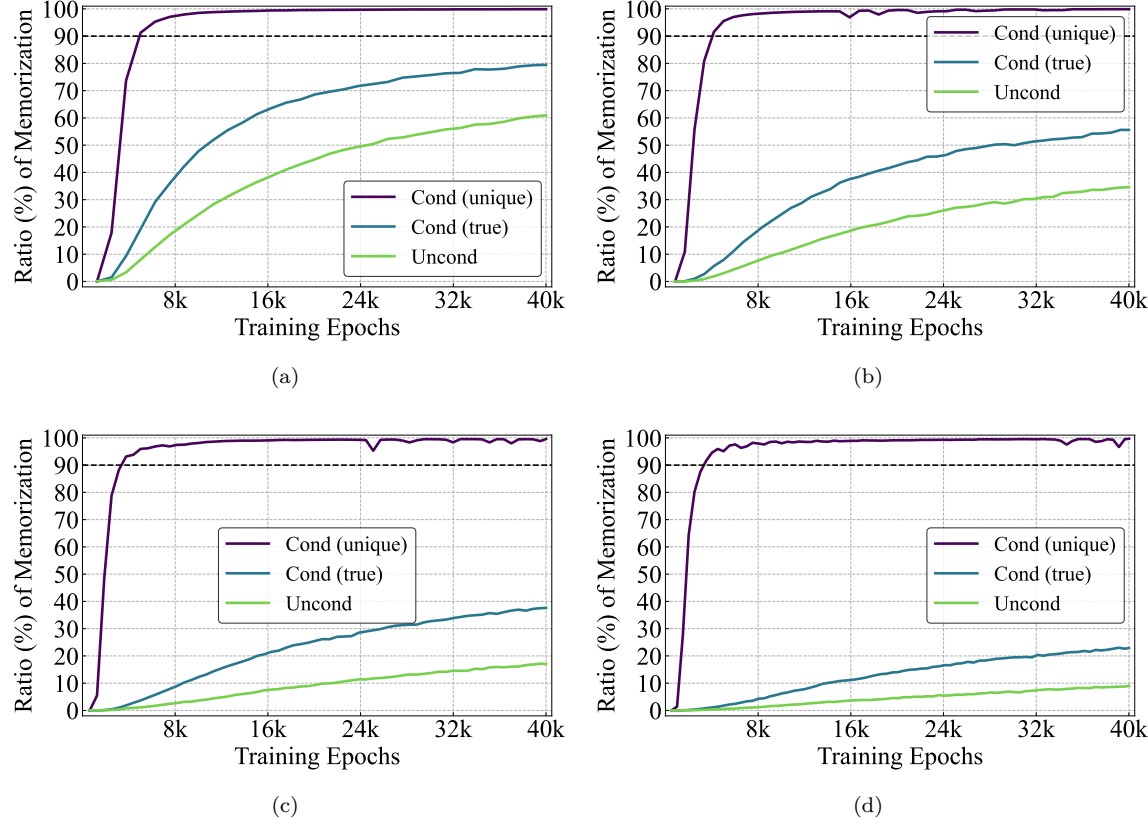

Figure 9: **Memorization ratios (%)** of unconditional / conditional (with true labels) / conditional (with unique labels) diffusion models on CIFAR-10 during the training at the training data size (a) $|\mathcal{D}| = 2k$; (b) $|\mathcal{D}| = 3k$; (c) $|\mathcal{D}| = 4k$; (d) $|\mathcal{D}| = 5k$.

## D.2 Time Embedding

We conduct experiments to compare two distinct time embedding methods within the model structure of DDPM++ (Song et al., 2021b; Karras et al., 2022): positional embeeding (Vaswani et al., 2017) and random fourier features (Tancik et al., 2020). As illustrated in Figure 11(b), by varying training data size $|\mathcal{D}|$, there

Table 4: **Memorization ratios (%)** of unconditional / conditional (true labels) / conditional (unique labels) diffusion models on CIFAR-10.

|  | $|\mathcal{D}| = 1k$ | $|\mathcal{D}| = 2k$ | $|\mathcal{D}| = 3k$ | $|\mathcal{D}| = 4k$ | $|\mathcal{D}| = 5k$ |
|---|---|---|---|---|---|
| Unconditional | 92.09 | 60.93 | 34.60 | 17.12 | 9.00 |
| Conditional (true labels) | 96.59 | 79.46 | 55.62 | 37.59 | 23.00 |
| Conditional (unique labels) | 100.00 | 99.88 | 99.89 | 99.57 | 99.66 |

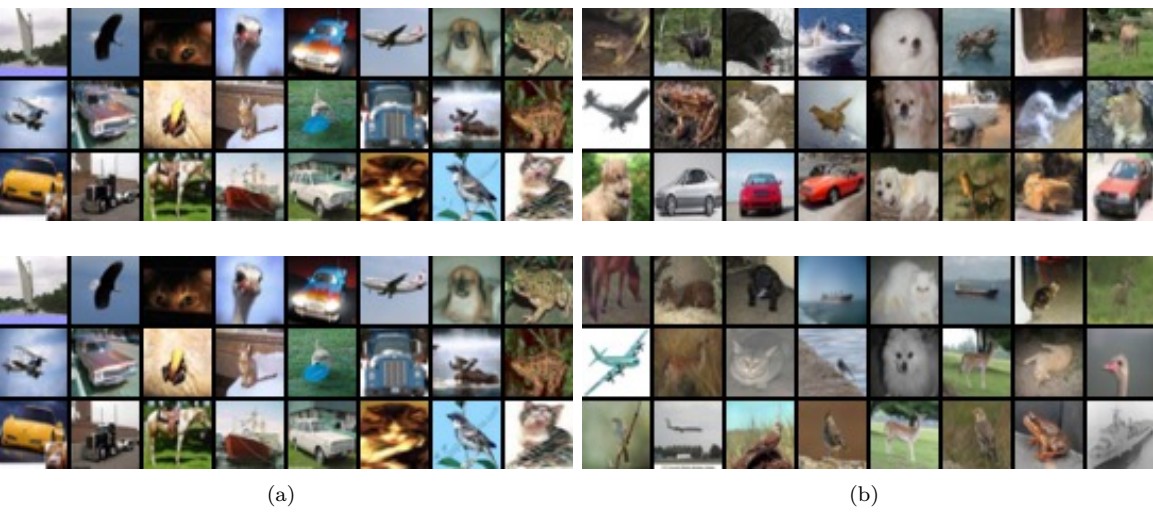

(a)                    (b)

Figure 10: Generated images (top three rows) and their $\ell_2$-nearest training samples in $\mathcal{D}$ (bottom three rows) by (a) the conditional EDM with unique labels (b) the unconditional EDM.

is notable decline in the memorization ratio when employing random fourier features for time embeddings in the DDPM++ model, which reconfirms our conclusions in the main paper.

### D.3 Unconditional v.s. Conditional Generation

In the main paper, we demonstrated the significant contributions of random labels to memorization within diffusion models. To further substantiate these findings, we conduct additional experiments using the FFHQ dataset. We note that FFHQ has no ground truth class labels for each image, so we only consider random labels and unique labels in our experimental design. Firstly, we construct a training dataset with $|\mathcal{D}| = 2k$. Subsequently, the number of classes $C$, for random labels, is varied within the set {1, 10, 50, 100, 200, 400, 2000}. Afterwards, we train conditional diffusion models on these training data of different random labels. As demonstrated in Figure 11(c), the memorization ratio for $C = 1$ (equivalent to the unconditional scenario) approximates to only 20%. However, it achieves over 90% with an increase in $C$ to 400. Additionally, we conduct another experiment to investigate the effects of unique labels. As visualized in Figure 11(d), when unique labels are provided as conditions to diffusion models, the memorization ratio still exceeds 90% at a training data size of $|\mathcal{D}| = 5k$. In contrast, for unconditional diffusion models, the memorization ratio diminishes even below 90% when $|\mathcal{D}| = 1k$. To summarize, these findings show that the number of classes for random labels plays a pivotal role in the memorization of diffusion models, thus aligning with our earlier results derived from experiments on the CIFAR-10 dataset.

## E Experimental Results of FIDs

In this section, we provide the Fréchet Inception Distances (FIDs) (Heusel et al., 2017) in parallel with the memorization ratios in the main paper. The results related to CIFAR-10 are shown in Figure (12-15) and

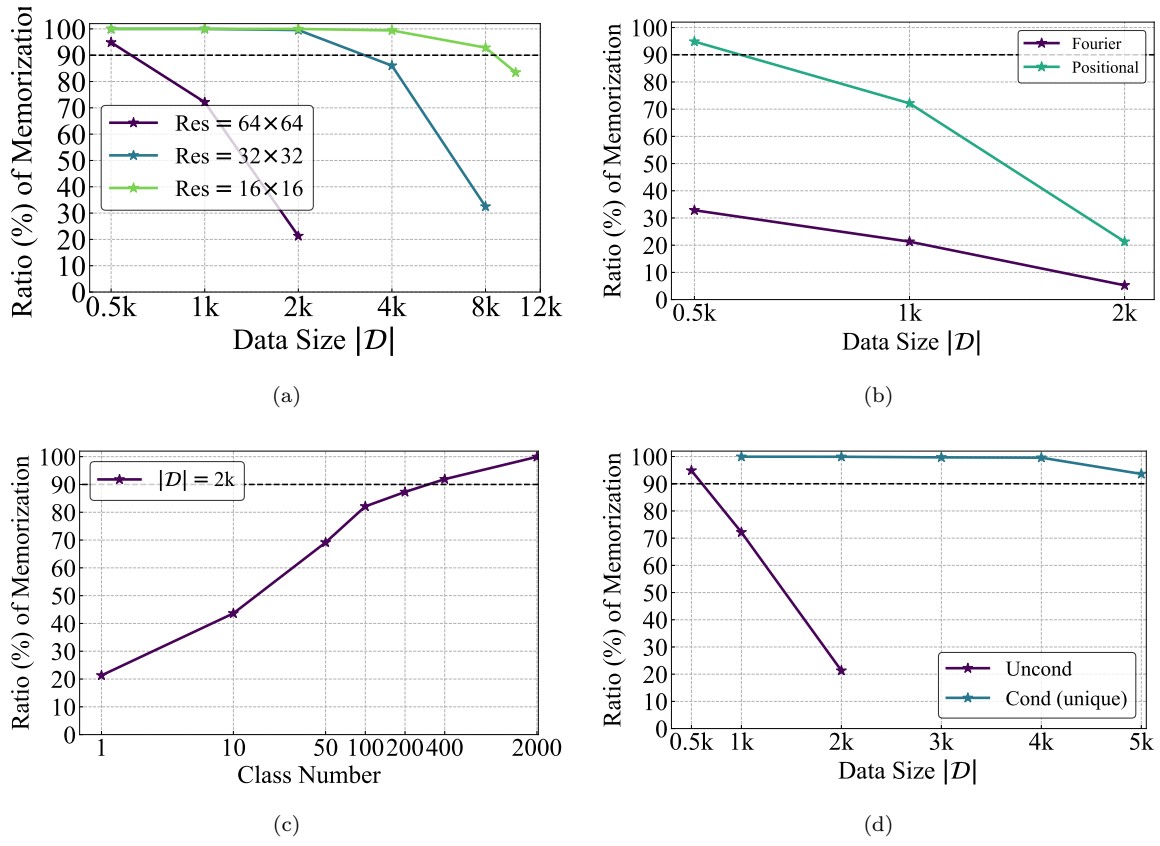

Figure 11: **Memorization ratio (%)** of diffusion models on FFHQ under (a) different data dimension; (b) different time embeddings; (c) different numbers of classes for random labels; (d) unconditional diffusion models and conditional diffusion models with unique conditions.

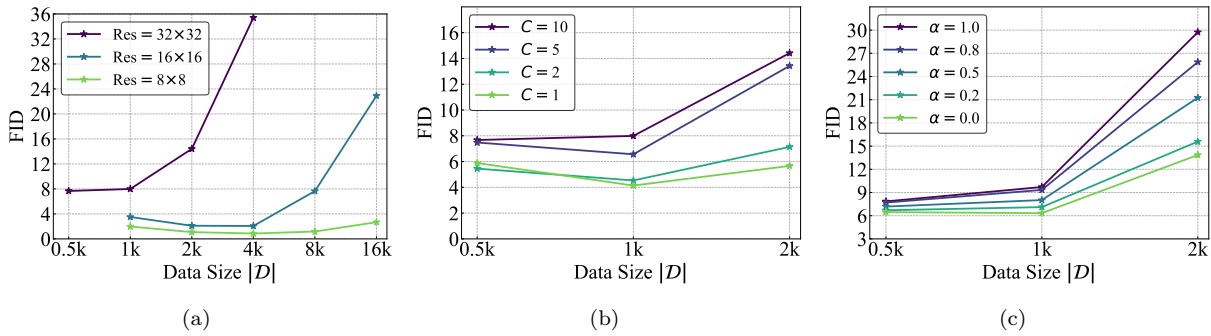

Figure 12: **FIDs** of diffusion models on CIFAR-10 under different (a) data dimensions; (b) inter-diversity; (c) intra-diversity.

Table (5-6). While the results on FFHQ are present in Figure 16. By analyzing the FID results, we observe that a configuration with higher memorization ratio tends to results in a lower FID. This observation is in line with the intuition: when diffusion models memorize a significant proportion of training data, FIDs are also close to optimal.

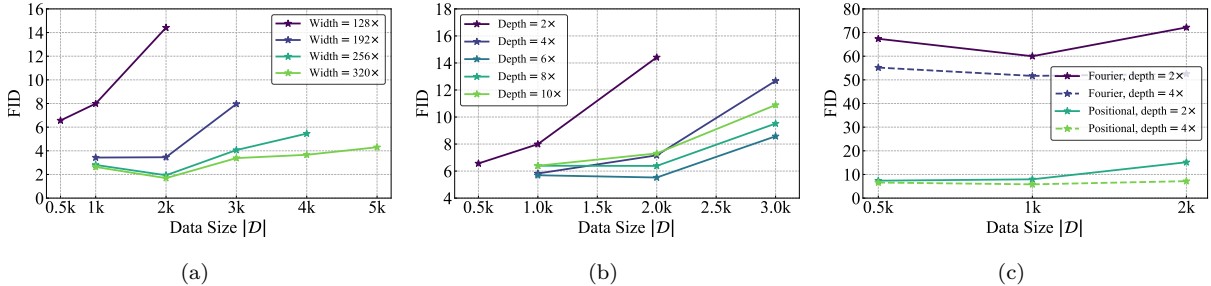

Figure 13: **FIDs** of diffusion models on CIFAR-10 under different (a) model widths; (b) model depths; (c) time embeddings.

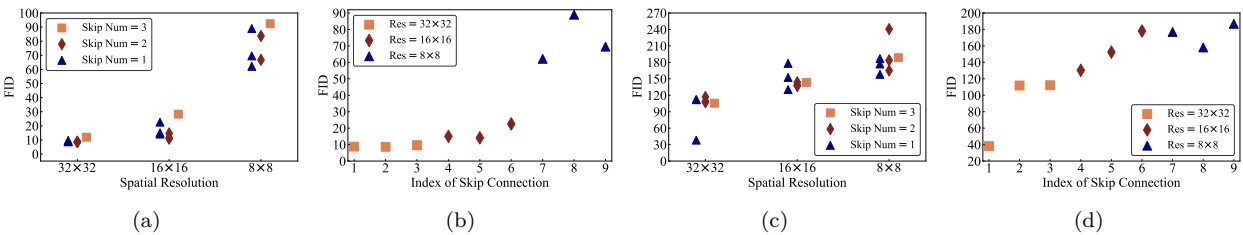

Figure 14: **FIDs** of diffusion models on CIFAR-10 when retaining (a) skip connections of certain spatial resolution for DDPM++; (b) single skip connection at different locations for DDPM++; (c) skip connections of certain spatial resolution for NCSN++; (d) single skip connection at different locations for NCSN++.

Table 5: **FIDs** of diffusion models on CIFAR-10 under different batch sizes.

| Batch Size | 128 | 256 | 384 | 512 | 640 | 768 | 896 |
|---|---|---|---|---|---|---|---|
| Learning rate $(10^{-4})$ | 0.5 | 1.0 | 1.5 | 2.0 | 2.5 | 3.0 | 3.5 |
| $\mathcal{R}_{\mathrm{mem}}(\%)$  $\quad |\mathcal{D}| =1\mathrm{k}$ | 8.89 | 8.60 | 8.14 | 7.99 | 8.25 | 7.98 | **7.85** |
| $\quad |\mathcal{D}| =2\mathrm{k}$ | 17.56 | 15.57 | 15.23 | **14.41** | 14.83 | 14.56 | 14.85 |

Table 6: **FIDs** of diffusion models on CIFAR-10 under different (*left*) weight decay values; (*right*) EMA values.

| Weight decay | FID $|\mathcal{D}| =1\mathrm{k}$ | FID $|\mathcal{D}| =2\mathrm{k}$ |
|---|---|---|
| $0$ | **7.99** | 14.41 |
| $1 \times 10^{-5}$ | 8.17 | 15.73 |
| $1 \times 10^{-4}$ | 8.08 | 14.36 |
| $1 \times 10^{-3}$ | 9.37 | **12.75** |
| $1 \times 10^{-2}$ | 13.47 | 17.02 |
| $5 \times 10^{-2}$ | 41.08 | 51.70 |
| $1 \times 10^{-1}$ | 63.78 | 48.21 |

| EMA | FID $|\mathcal{D}| =1\mathrm{k}$ | FID $|\mathcal{D}| =2\mathrm{k}$ |
|---|---|---|
| 0.99929 | **7.99** | **14.41** |
| 0.999 | 8.11 | 15.75 |
| 0.99 | 8.55 | 17.11 |
| 0.9 | 9.83 | 18.04 |
| 0.5 | 10.32 | 21.28 |
| 0.1 | 12.38 | 23.12 |
| 0.0 | 11.77 | 22.39 |

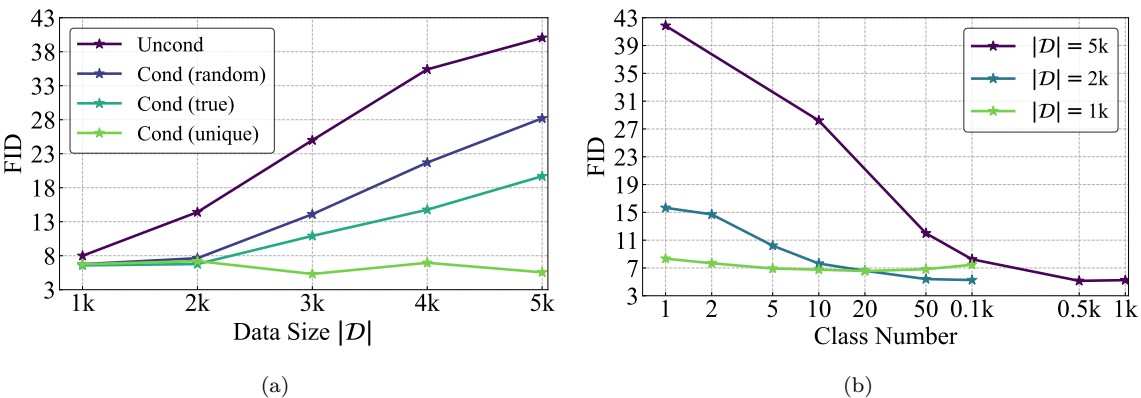

Figure 15: **FIDs** of (a) unconditional diffusion models and conditional diffusion models with true / random / unique labels on CIFAR-10; (b) conditional diffusion models with random labels of different $C$.

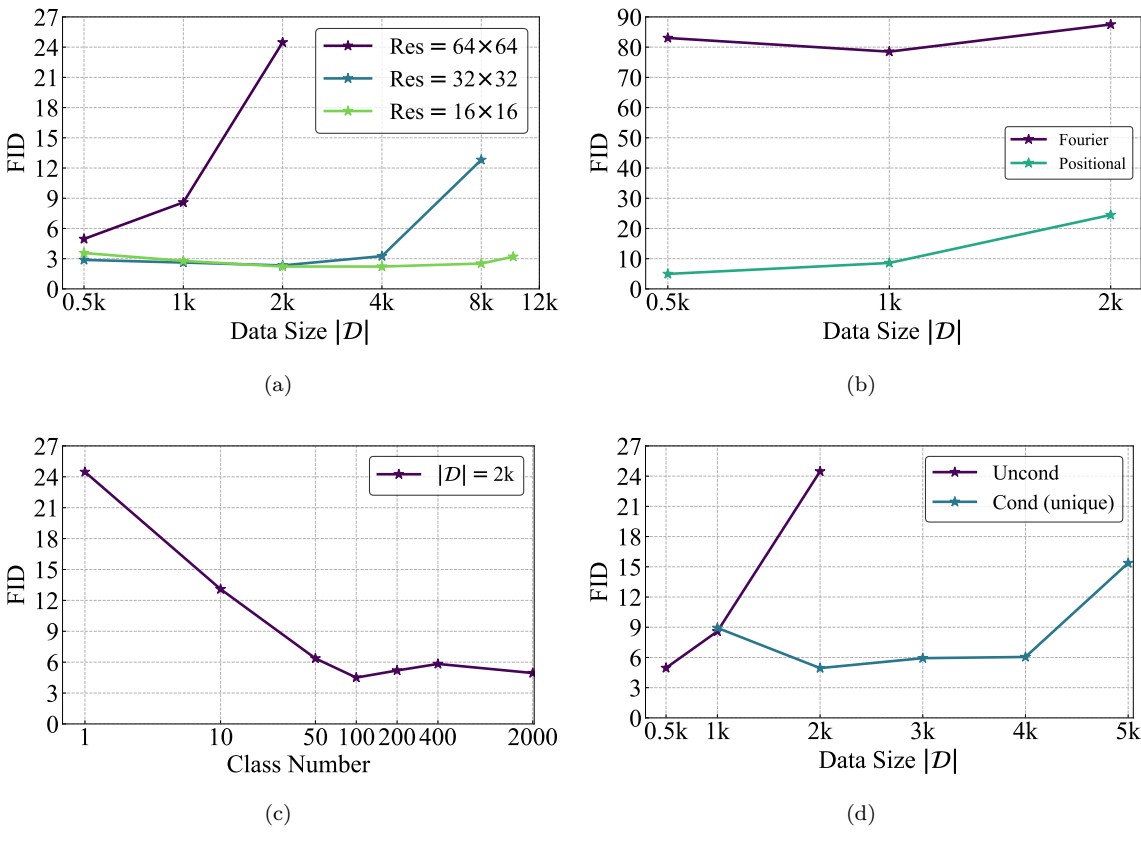

Figure 16: **FIDs** of diffusion models on FFHQ under (a) different data dimension; (b) different time embeddings; (c) different numbers of classes for random labels; (d) unconditional diffusion models and conditional diffusion models with unique conditions.

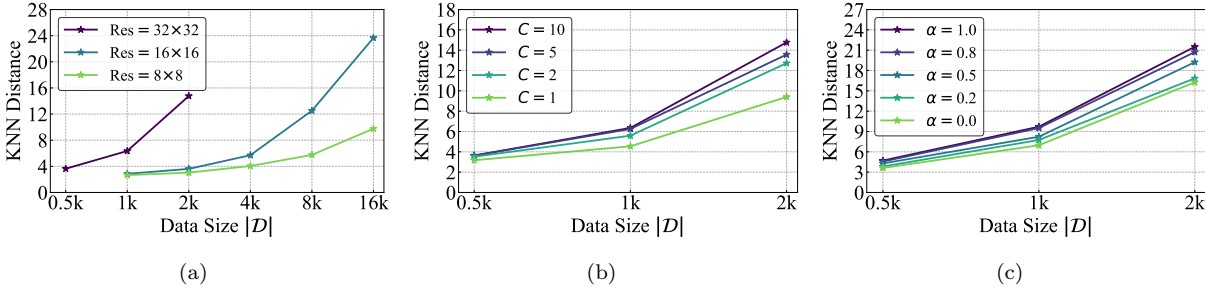

Figure 17: **KNN distances** of diffusion models on CIFAR-10 under different (a) data dimensions; (b) inter-diversity; (c) intra-diversity.

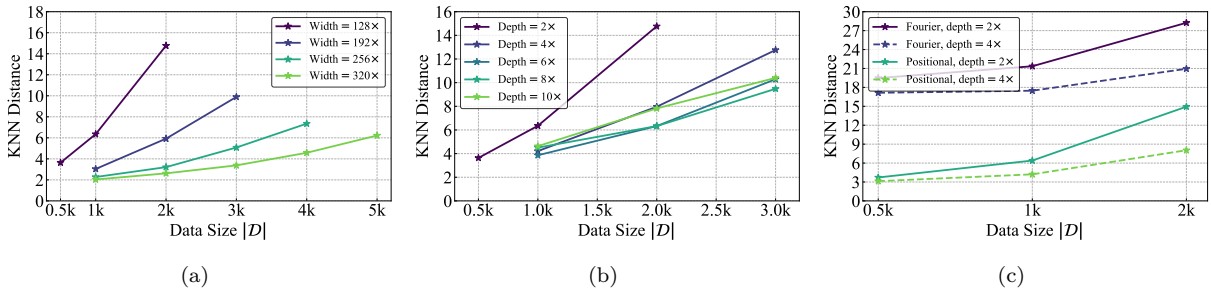

Figure 18: **KNN distances** of diffusion models on CIFAR-10 under different (a) model widths; (b) model depths; (c) time embeddings.

## F  Experimental Results of Alternative Memorization Criteria

In this section, we aim to corroborate our findings regarding memorization of diffusion models by incorporating alternative memorization metrics. The memorization ratio, as defined in the main paper, can be formally articulated as follows

$$\mathcal{R}_{\text{Mem}} = \frac{1}{N'} \sum_{n=1}^{N'} \mathbb{I}\big(\frac{\|x'_n, \text{NN}_1(x'_n, \mathcal{D})\|_2}{\|x'_n, \text{NN}_2(x'_n, \mathcal{D})\|_2} < \frac{1}{3}\big), \tag{62}$$

where $\mathbb{I}$ is an indicator function, $\text{NN}_j(x'_n, \mathcal{D})$ is the $j$-th nearest training sample of $x'_n$ in $\mathcal{D}$, and $N'$ is the number of generations sampled by the diffusion model.

We also consider the KNN distance used in Carlini et al. (2023) as a surrogate memorization metric:

$$\mathcal{R}'_{\text{Mem}} = \frac{1}{N'} \sum_{n=1}^{N'} \|x'_n, \text{NN}_1(x'_n, \mathcal{D})\|_2. \tag{63}$$

It is noticed that when memorization ratio is high or KNN distance is low, the diffusion models feature memorizing more on the training data. Afterwards, we re-evaluate our results in the main paper by using the KNN distance as memorization metric. The results on CIFAR-10 are shown in Figure (17-20) and Table (7-8) while the results related to FFHQ are present in Figure 21. We notice that these new results are in alignment with our original conclusions using the memorization ratio metric.

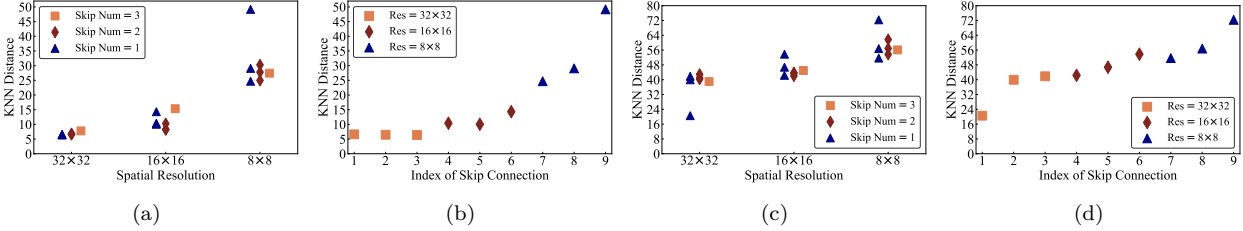

Figure 19: **KNN distances** of diffusion models on CIFAR-10 when retaining (a) skip connections of certain spatial resolution for DDPM++; (b) single skip connection at different locations for DDPM++; (c) skip connections of certain spatial resolution for NCSN++; (d) single skip connection at different locations for NCSN++.

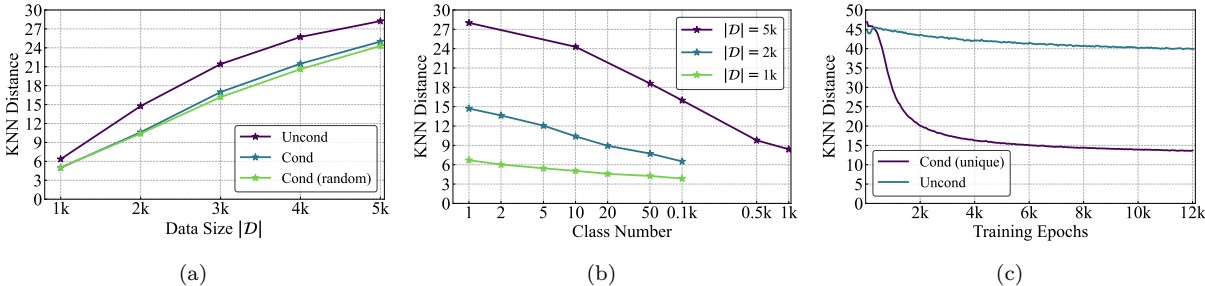

Figure 20: **KNN distances** of (a) unconditional diffusion models and conditional diffusion models with true labels or random labels on CIFAR-10; (b) conditional diffusion models with random labels of different $C$; (c) conditional EDM with unique labels and unconditional EDM at $|\mathcal{D}| = 50$k during the training.

Table 7: **KNN distances** of diffusion models on CIFAR-10 under different batch sizes.

| Batch Size | 128 | 256 | 384 | 512 | 640 | 768 | 896 |
|---|---|---|---|---|---|---|---|
| Learning rate $(10^{-4})$ | 0.5 | 1.0 | 1.5 | 2.0 | 2.5 | 3.0 | 3.5 |
| KNN distance $\quad |\mathcal{D}| = 1$k | 7.35 | 6.73 | 6.51 | 6.35 | 6.51 | 6.40 | **6.30** |
| $\qquad\qquad\qquad |\mathcal{D}| = 2$k | 15.43 | 14.92 | 14.88 | 14.77 | 14.53 | **14.12** | 14.57 |

Table 8: **KNN distances** of diffusion models on CIFAR-10 under different (*left*) weight decay values; (*right*) EMA values.

| Weight decay | KNN distance $|\mathcal{D}| = 1$k | $|\mathcal{D}| = 2$k |
|---|---|---|
| 0 | **6.35** | 14.77 |
| $1 \times 10^{-5}$ | 6.45 | **14.63** |
| $1 \times 10^{-4}$ | 6.42 | 15.29 |
| $1 \times 10^{-3}$ | 7.32 | 16.99 |
| $1 \times 10^{-2}$ | 9.99 | 23.40 |
| $5 \times 10^{-2}$ | 26.69 | 38.73 |
| $1 \times 10^{-1}$ | 34.44 | 39.51 |

| EMA | KNN distance $|\mathcal{D}| = 1$k | $|\mathcal{D}| = 2$k |
|---|---|---|
| 0.99929 | **6.35** | 14.77 |
| 0.999 | 6.44 | **14.58** |
| 0.99 | 6.51 | 15.21 |
| 0.9 | 7.02 | 15.39 |
| 0.5 | 7.11 | 15.04 |
| 0.1 | 6.90 | 15.08 |
| 0.0 | 7.08 | 14.59 |

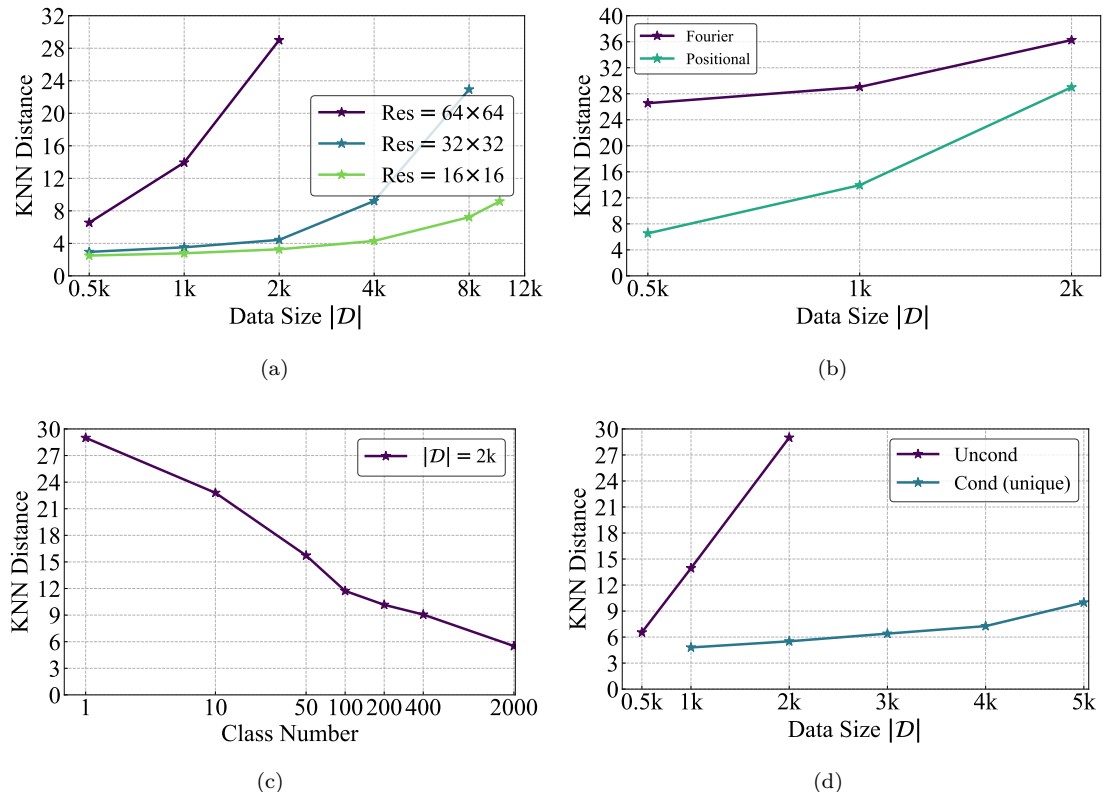

Figure 21: **KNN distances** of diffusion models on FFHQ under (a) different data dimension; (b) different time embeddings; (c) different numbers of classes for random labels; (d) unconditional diffusion models and conditional diffusion models with unique conditions.

## G    Implementation Details of Finetuning on Stable Diffusion

**Data distribution.** Imagenette consists of $C = 10$ classes from the ImageNet dataset (Deng et al., 2009). The classes include "chain saw", "garbage truck", "tench", "French horn", "gas pump", "English springer", "parachute", "church", "cassette player", "golf ball". In our experiments, the image resolution is specified as $256 \times 256$. Similar to our experiments on CIFAR-10 and FFHQ, we disable all data augmentation.

**Model configuration.** Stable diffusion (Rombach et al., 2022) includes an image encoder, an image decoder, a unet, and a text encoder. The image encoder is used to encode images into latent representations while the decoder reconstructs images from the latent. The unet plays a role as the latent diffusion model conditioning on text embeddings. Before fine-tuning the model on the Artbench dataset, we load the model weights from pre-trained stable diffusion[4].

**Training procedure.** During the fine-tuning, we only train the unet in stable diffusion and freeze the model weights of image encoder/decoder and text encoder. We adopt a learning rate of $1 \times 10^{-4}$ and $3 \times 10^{-4}$ for full fine-tuning and LoRA fine-tuning (Hu et al., 2021) with cosine learning rate schedule and weight decay of $1 \times 10^{-6}$. The batch size is set as 16 and the gradient accumulation step is set as 4. For each training dataset $\mathcal{D}$, we enable the EMA and fine-tune the stable diffusion until 400k images have been observed.

**Text-to-image generation.** We follow the pipeline of DDIM (Song et al., 2021a) with 50 steps to sample 10k images from fine-tuned stable diffusion. During the generation, we disable the safety checker to prevent generating black images. For *plain conditioning*, we use the text prompt "a picture". While for *class conditioning*, we sample 1k images using the prompt "a picture of <CLASS_NAME>" for each class, totaling 10k generated samples.

---

[4]https://huggingface.co/lambdalabs/miniSD-diffusers

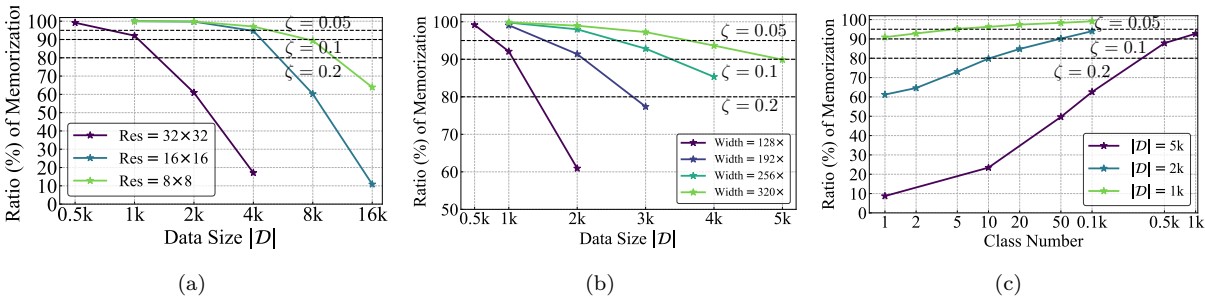

Figure 22: EMMs of diffusion models with various $\zeta$ under configurations of (a) different data dimensions; (b) different model widths; (c) different numbers of classes for random labels.

# H Sensitivity of Effective Model Memorization on $\zeta$

The EMM metric defined in 2.1 is used to quantify the conditions which empirically trained diffusion models exhibit memorization behavior similar to the theoretical optimum. The selection of $\zeta$ determines the closeness between the empirical model and theoretical one. Generally, a smaller $\zeta$ requires a higher memorization ratio for the empirical diffusion model to approximate the theoretical optimum. In Figure 22, we select various $\zeta$ in different configurations. We observe that a smaller $\zeta$ typically results in a smaller EMM. Meanwhile, the conclusions in our main paper remain consistent. For instance, smaller data resolutions, larger model widths, larger numbers of classes for conditioning lead to more memorization.

