# OpenReview forum: "On Memorization in Diffusion Models"
_TMLR — Accepted by TMLR_

### Review · Reviewer_bw9d · 2024-11-26

**Summary Of Contributions:**

The paper examines the phenomenon of memorization in diffusion models. By defining Effective Model Memorization (EMM), the authors systematically study the factors that influence memorization, including data distribution, model configuration, and training procedures. Key contributions include:
* Establishing the theoretical optimum of diffusion models and highlighting its tendency to memorize training data.
* Empirical insights into how dataset size, model architecture, and training conditions affect memorization, with CIFAR-10 as a primary dataset.
* Surprising observations, such as the role of uninformative random labels in triggering memorization.
* Analysis of memorization across various models and setups.

**Audience:**

Yes

**Claims And Evidence:**

Yes

**Requested Changes:**

1. Clarify Theoretical Foundations: Elaborate on why Eq. (5) is a convex optimization problem. Please, provide specific conditions or assumptions that justify this claims.

2. State Assumptions Explicitly: Clearly outline the assumptions underlying the theoretical results.

3. Explain which assumptions lead to deviations between empirical and theoretical results. Specifically, what implicit or explicit assumption removes the effect of sample size (and other factors such as steps, training time etc.) on memorization. Why does empirical results diverge from theory? Is the cause primarily due to model capacity, dataset characteristics or some other factor? A brief discussion would help readers.

4. Extend Beyond CIFAR-10 (optional): Include experiments on larger and more diverse datasets (e.g., ImageNet) to validate the generalizability of the findings.

**Strengths And Weaknesses:**

Strengths:

* Novel Metric for Memorization: The introduction of EMM provides a concrete way to quantify and explore memorization across different setups.
* Comprehensive Experiments: The paper evaluates multiple factors, including data diversity, resolution, model size, and conditional labels, providing robust empirical support for its claims.
* Practical Implications: Insights into privacy risks and training considerations in generative models are relevant for real-world applications.

Weaknesses:

*  Convexity Argument Ambiguity: The claim that Eq (5) is a convex optimization problem requires elaboration, especially since $s_\theta$​ seems inherently non-convex in complex architectures. This theoretical foundation needs strengthening.
* Assumptions: The assumptions underlying the theoretical results are not clearly stated, making it difficult to reconcile the theory with empirical observations.
*  Limited Dataset Variety: Over-reliance on CIFAR-10 may restrict generalizability due to its low resolution and limited size, especially when larger datasets are essential for practical use cases.

---

> ### Author Response · Authors · 2024-12-24
>
> Thank you for your supportive review and suggestions. Below we respond to the comments in **Weaknesses (W)** and **Requested Changes (R)**.
>
> ---
>
> ***W1 and R1: Convexity Argument Ambiguity in Eq (5). The conditions or assumptions for the claim.***
>
> Thank you for pointing this out. We would like to clarify that the objective in Eq. (5) is convex w.r.t. the score $\\boldsymbol{s}\_{\\theta}$. The analysis in Appendix A.1 focuses on the score $\\boldsymbol{s}\_{\\theta}$ instead of the parameters $\\theta$ (as can be observed from the fact that we take the gradient w.r.t. $\\boldsymbol{s}\_{\\theta}$ to derive the optimal solution). Indeed, when $\\boldsymbol{s}\_{\\theta}$ is parameterized using a neural network $\\theta$, the objective in Eq. (5) is typically highly non-convex w.r.t. $\\theta$. We have clarified this distinction in $\\textrm{\\color{blue}Appendix A.1}$ in the revision.
>
> ---
>
> ***W2 and R2 and R3: Assumptions underlying the theoretical results. Discussions on the deviations between empirical and theoretical results.***
>
> In deriving the optimal diffusion model, we assume that the training dataset consists of a finite number of samples, such that the empirical data distribution can be represented using delta functions $\\widehat{\\mathbf{P}}(x)=\\frac{1}{N}\\sum\_{n=1}\^N\\delta(x-x\_n)$. This assumption is widely applicable when training diffusion models on a given dataset. As long as the sample size $N$ is finite, we can obtain the theoretical optimum, which only replicates/memorizes the training data.
>
> However, achieving this theoretical optimum necessitates a model $\\theta$ with sufficient capacity.  When $\\boldsymbol{s}\_{\\theta}$ is parameterized by a model family $\\theta$ with finite capacity, e.g. UNet, the theoretical optimum may lie outside its solution space. Even if the solution space of $\\theta$ encompasses the theoretical optimum, reaching it remains challenging due to the highly non-convex nature of the objective in Eq (5) w.r.t. model parameters $\\theta$.
>
> We observe that the empirical diffusion models do not exhibit behavior resembling the theoretical optimum, as they do not merely memorize the training data. This discrepancy motivates us to investigate how factors such as data distribution (including sample size and dataset characteristics), model configuration (e.g., capacity), and the training process (e.g., optimization setup and training duration) contribute to these deviations.
>
> ---
>
> ***W3 and R4: Experiments beyond CIFAR-10 to show the generalizibility of conclusions***
>
> Thank you for your feedback. Besides the experiments on CIFAR-10 presented in the main paper, we conducted further experiments on the FFHQ dataset, as detailed in $\\textrm{\\color{blue}Appendix D}$. Specifically, we explored the impact of data dimension, time embedding, and the number of class conditions on memorization. The results consistently support the conclusions drawn in the main paper. Furthermore, in $\\textrm{\\color{blue}Section 6}$, we fine-tuned Stable Diffusion on a subset of the ImageNet dataset to further demonstrate the generalizability of our conclusions.

---

> > ### Comment · Reviewer_bw9d · 2025-01-14
> >
> > Thank you for your response, I think this clarifies my questions.

---

> > > ### Author Response · Authors · 2025-01-14
> > >
> > > We appreciate your detailed feedback and suggestions, which greatly help us to improve our work!

---

### Review · Reviewer_td2T · 2024-11-29

**Summary Of Contributions:**

This paper is on memorization in diffusion models and conducts an ablation study as a function of training-data, model-sizes, optimization-configurations and conditioning. To that end, the paper introduces a metric called "effective-model memorization" that is defined as the number of training samples for which the model does not generalize beyond the empirical score-estimate (i.e., a non-parametric score-estimator derived using samples in the training-dataset).

**Audience:**

Yes

**Claims And Evidence:**

Yes

**Requested Changes:**

I do not have strong recommendations for revisions. But, I would appreciate it if the authors could address the weaknesses --- for e.g., by pointing out findings that exclusively hold for diffusion-models but not other standard classification/generative models.

**Strengths And Weaknesses:**

**Strengths:** The paper contains extensive ablation studies and their analysis includes simpler unconditional diffusion-models as well as the larger text-conditional diffusion models.

**Weaknesses:**
1. In my opinion, the major weakness is that most of these results are expected. For e.g., it is indeed natural and even expected to observe memorization with reduced training datasets. From this perspective, it may seem like the contributions of the paper are very limited. However, the extensive ablation study and experimental findings may be of some interest to future practitioners in this area.
2. The paper appears to suggest that memorization in diffusion models is solely due to the fact that a closed-form solution of optimal score-function exists. I disagree with this characterization. For e.g., consider normalizing-flow: while the theoretically-optimal flow transformation for a small set of images is not immediately clear, it would almost surely overfit given sufficient capacity. In fact, many findings in this work are not specific to diffusion-model training and are a direct consequence of the supervised-training.

Overall, I do not strongly feel against this paper since the experimental analysis and results are still interesting to know (even if many of these are expected). I am interested to see what other reviewers/AE feel about the paper.

---

> ### Author Response · Authors · 2024-12-24
>
> Thank you for your supportive review and suggestions. Below we respond to the comments in **Weaknesses (W)** and **Requested Changes (R)**.
>
> ---
>
> ***W1: The major weakness is that most of these results are expected.***
>
> The interaction between data and model capacity has long been a key topic for both discriminative and generative models. Accordingly, most of our experiments on model and data could be summarized as: “less training data or more model capacity increases memorization”. This observation aligns with the expectations from prior literature. We believe that the extensive ablation studies we conducted, as you mentioned, could serve as guidelines for practitioners working with diffusion models. In particular, the effects of model depth, skip connections, EMA on memorization may not be immediately intuitive.
>
> Moreover, the effects of conditioning, as highlighted in our paper, exceed our expectations. From [1] and our experiments, unconditional diffusion models trained on the full set of CIFAR-10, or even larger training data, exhibit near-zero memorization ratios. However, with much more diverse conditioning, the resulting diffusion models can memorize a significant proportion of training data. This explains why Stable Diffusion, trained on millions of images, still displays notable memorization behavior.
>
> ---
>
> ***W2: The paper appears to suggest that memorization in diffusion is solely due to the fact that a closed-form solution of optimal score-function exists … many findings in this work are not specific to diffusion-model training and are a direct consequence of the supervised-training***
>
> Thank you for your comments and suggestions. We would like to clarify our motivations. In $\\textrm{\\color{blue}Appendix A.1}$, we demonstrate that there exists an optimal solution for the denoising score matching objective, and diffusion models are trained to approximate such a theoretical optimum, as described in Equation 22. Although this theoretical optimum can only memorize training data (from both empirical and theoretical perspectives), the empirically trained diffusion models are capable of generating novel samples. This intriguing gap motivates us to investigate the factors in the diffusion model training recipe that prevent diffusion models from converging to the theoretical optimum.
>
> We recognize that with limited training data, generative models (not limited to diffusion models) tend to overfit the empirical data distribution: $\\widehat{\\mathbf{P}}(x)=\\frac{1}{N}\\sum\_{n=1}\^N\\delta(x-x\_n)$, leading to the memorization of training data. Compared to other generative models, such as VAEs and GANs, diffusion models have an explicitly defined optimal solution (Equation 2), which provides a unique perspective for understanding memorization and generalization in these models.
>
> We believe our extensive explorations would be helpful for practitioners to better understand the gap between empirical diffusion models and the theoretical optimum. While our focus is on diffusion models, many factors in the diffusion model training recipe (excluding residual connections in U-Net and EMA values) may also apply to other generative models. It would be valuable to explore whether our conclusions hold for other generative models, particularly regarding the impact of conditioning.
>
> ---
>
> ***References:*** \
> [1] Carlini et al. Extracting training data from diffusion models. USENIX 2023

---

> > ### Comment · Reviewer_td2T · 2025-01-14
> > **Thank you for your response!**
> >
> > Dear authors,
> >
> > I apologize for the delayed response! Thank you very much for your clarifications: I recognize that diffusion models are uniquely situated in terms of optimal solution being available but at the same time, I still feel that most of the observations are predictable! I vote for acceptance since the ablations conducted in this paper are valuable for researchers in this area (including myself).
> >
> > I also would like to point the authors to this recent related preprint: https://arxiv.org/abs/2412.20292. I request the authors to consider including a small discussion on this paper.
> >
> > Thank You,
> > Sincerely,
> > Reviewer td2T.

---

> ### Author Response · Authors · 2025-01-14
>
> Thank you for your supportive comments. We also noticed the paper [1] you mentioned, which shares motivations similar to ours: diffusion models trained on finite datasets are expected to learn the ideal score function that merely memorizes the training data—raising the question of why empirically trained models can generate novel samples.
>
> In [1], the authors identify two inductive biases (locality and equivariance) in common architectures of diffusion models that prevent convergence to the ideal score function. Building on this insight, they theoretically derive equivariant local score (ELS) machines, which mirror the formulation of the ideal score function while exhibiting the generalization behavior of empirically trained models. However, these two inductive biases do not fully capture the behavior of trained models, particularly those with self-attention architectures. Analytical characterization of the exact properties of trained diffusion models remains challenging.
>
> Our work, on the empirical side, investigates how data, model designs, training processes, and conditioning prevent trained models from converging to the ideal score function. While our primary focus is on the memorization phenomenon, we believe our empirical study also provides valuable insights into the inductive biases that contribute to generalization in trained diffusion models.
>
> We have incorporated a similar discussion in our revision in $\\textrm{\\color{blue}Section 7}$.
>
> ---
>
> ***References:*** \
> [1] Kamb et al. An analytic theory of creativity in convolutional diffusion models. Arxiv 2024.

---

### Review · Reviewer_xsGH · 2024-12-02

**Summary Of Contributions:**

This paper studies memorization in diffusion models, introducing the Effective Model Memorization (EMM) metric to quantify conditions that lead to memorization. Experiments show that smaller datasets and random label conditioning significantly increase memorization rates, highlighting privacy risks and offering insights for improving generative model design.

**Audience:**

Yes

**Claims And Evidence:**

Yes

**Requested Changes:**

1. **Refine Metric Justifications**: Provide a detailed rationale for chosen thresholds, such as the 90% memorization ratio, and explore the impact of varying these values.

2. **Discussion on real-world scenarious**: the experiments are conducted on very small datasets, raising questions about how these findings translate to real-world scenarios involving large-scale datasets and models trained on millions of samples. Adding a discussion on the scalability and applicability of the results to such scenarios would provide valuable context and strengthen the study's relevance to practical applications.

**Strengths And Weaknesses:**

### Strengths
1. **Significant Advancement in Diffusion Models**: Comprehensive exploration of diffusion models, addressing their capability for high-quality generative tasks across diverse domains such as image, audio, and graph generation.
2. **Empirical and Theoretical Analysis**: Strong empirical results combined with a robust theoretical foundation, including the derivation of a closed-form solution for the optimal score model.
3. **Comprehensive Factor Analysis**: Detailed investigation into data distribution, model configuration, and training procedures, and their influence on memorization behaviors.

### Weaknesses
1. **Focus on Small Datasets**: The study places significant emphasis on smaller datasets, which may constrain the generalizability of its findings when applied to larger, production-scale datasets commonly used in diffusion models. Specifically, the experimental setup is simplified by employing a subset of the CIFAR-10 dataset, limiting its applicability to more complex and diverse real-world scenarios.
2. **Focus only on single metric**:  The results lack a discussion of other relevant metrics, such as the quality of generated outputs or the convergence of the loss function.

---

> ### Author Response · Authors · 2024-12-24
>
> Thank you for your supportive review and suggestions. Below we respond to the comments in **Weaknesses (W)** and **Requested Changes (R)**.
>
> ---
> ***W1: Focus on Small Datasets.***
>
> Our primary goal is to understand (how various factors influence) the gap between an empirically trained diffusion model and the theoretical optimum, as the former is theoretically expected to approximate the latter. We conduct experiments to observe the effect of training data size on memorization, as shown in $\\textrm{\\color{blue}Figure 1}$. When using the full CIFAR-10 dataset for training, the memorization ratio is close to zero. This observation aligns with the results in [1], which show that only 200-300 memorized training images are extracted from $2\^{20}$ generated images by DDPM and its variant trained on CIFAR-10. However, when the training data size is reduced, the resulting diffusion models exhibit more pronounced memorization. Since our focus is on investigating the regime where diffusion models approximate the theoretical optimum, we use subsets of CIFAR-10 rather than the full dataset in our experiments.
>
> Besides the CIFAR-10 experiments presented in the main paper, we also conduct experiments on FFHQ in $\\textrm{\\color{blue}Appendix D}$ and ImageNet in $\\textrm{\\color{blue}Section 6}$. These datasets are typically more complex and diverse than CIFAR-10. Nevertheless, our conclusions remain consistent across these datasets.
>
> ---
>
> ***W2: Focus only on single metric.***
>
> Besides the memorization ratio discussed in the main paper, we also consider other relevant metrics, such as FIDs in $\\textrm{\\color{blue}Appendix E}$ and KNN distances in $\\textrm{\\color{blue}Appendix F}$.
>
> We note that when diffusion models approximate the theoretical optimum, i.e.,  memorizing a significant proportion of training data, the quality metrics (e.g., FID) are also close to optimal. This intuition aligns with our findings in $\\textrm{\\color{blue}Appendix E}$, where configurations with higher memorization ratios tend to achieve lower FIDs. Additionally, while EMA values contribute limitedly to memorization, appropriate selections can lead to improved FIDs, as demonstrated in $\\textrm{\\color{blue}Table 6}$.
>
> We also explore alternative memorization metrics by following [1] and adopt the KNN distance between the generated images and their nearest training samples in $\\textrm{\\color{blue}Appendix F}$. After re-evaluating our experimental results, these new observations remain consistent with our original conclusions.
>
> ---
>
> ***R1: Refine Metric Justifications.***
>
> Our EMM metric is defined to quantify the conditions in which empirically trained diffusion models exhibit memorization behavior similar to the theoretical optimum. The threshold $1-\\zeta$ determines the degree of closeness between the empirical model and theoretical one. Consequently, there is no principled rule for selecting the value of $\\zeta$, as long as $\\zeta$ is not too large. Generally, a smaller $\\zeta$ requires a higher memorization ratio for the empirical diffusion model to approximate the theoretical optimum. Based on our experiments in $\\textrm{\\color{blue}Figure 1(d)}$, we set $\\zeta=0.1$.
>
> We further evaluate the experimental results under different values of $\\zeta$, including 0.05 and 0.2. Our findings indicate that a smaller $\\zeta$ results in a smaller EMM, meanwhile the conclusions are not sensitive to its choices. We have included this in $\\textrm{\\color{blue}Appendix H}$ in the revision.
>
> ---
>
> ***R2: Discussion on real-world scenarios.***
>
> We clarify the motivations for using small subsets in the response to ***W1***. Based on this understanding, when scaling unconditional diffusion models to millions of training samples, we expect the trained models to exhibit a near-zero memorization ratio. To further explore the memorization behavior of text-conditioned diffusion models trained on large-scale data, we conduct experiments on conditional diffusion models in $\\textrm{\\color{blue}Section 6}$. These experiments reveal that a larger number of class conditions, i.e., greater diversity of conditions during training, significantly triggers the memorization of diffusion models, even with large-scale training data. This explains why Stable Diffusion (SD), unlike unconditional diffusion models, memorizes a substantial proportion of training data.
>
> Since training SD from scratch under various configurations is intractable, we instead investigate a realistic scenario where SD is fine-tuned on customized data. In this setup, we examine the effects of customized data size, LoRA rank, and diversity of prompts on memorization. Our findings indicate that fine-tuned Stable Diffusion tends to memorize training data under smaller data size, more model capacity, and more diverse text prompts.
>
> ---
>
> ***References:*** \
> [1] Carlini et al. Extracting training data from diffusion models. USENIX 2023

---

### Decision · Action_Editor_x65q · 2025-01-23

**Recommendation:** Accept as is

**Comment:**

Overall, the main contribution of this paper is to provide a nice principled memorization method for diffusion models. The claims of the paper seem to be well supported and the reviewers are supportive of acceptance.

**Audience:**

the trustworthy AI community

**Claims And Evidence:**

Overall, the main contribution of this paper is to provide a nice principled memorization method for diffusion models. The claims of the paper seem to be well supported and the reviewers are supportive of acceptance.

---

> ### Author Response · Authors · 2025-02-15
>
> Dear Action Editor,
>
> Thank you for your supportive comments. We have revised our paper according to the comments from the reviewers and uploaded the camera-ready version of our paper.
>
> Sincerely,
>
> The Authors